# mTOR signalling controls the formation of smooth muscle cell-derived luminal myofibroblasts during vasculitis

Angus T Stock [1✉], Sarah Parsons [2,3], Jacinta A Hansen[1], Damian B D'Silva[1], Graham Starkey [4,5], Aly Fayed [6], Xin Yi Lim [7], Rohit D'Costa [8,9], Claire L Gordon [7,10,11] & Ian P Wicks [1,12,13✉]

## Abstract

The accumulation of myofibroblasts within the intimal layer of inflamed blood vessels is a potentially catastrophic complication of vasculitis, which can lead to arterial stenosis and ischaemia. In this study, we have investigated how these luminal myofibroblasts develop during Kawasaki disease (KD), a paediatric vasculitis typically involving the coronary arteries. By performing lineage tracing studies in a murine model of KD, we reveal that luminal myofibroblasts develop independently of adventitial fibroblasts and endothelial cells, and instead derive from smooth muscle cells (SMCs). Notably, the emergence of SMC-derived luminal myofibroblasts—in both mice and patients with KD, Takayasu's arteritis and Giant Cell arteritis—coincided with activation of the mechanistic target of rapamycin (mTOR) signalling pathway. Moreover, SMC-specific deletion of mTOR signalling, or pharmacological inhibition, abrogated the emergence of luminal myofibroblasts. Thus, mTOR is an intrinsic and essential regulator of luminal myofibroblast formation that is activated in vasculitis patients and therapeutically tractable. These findings provide molecular insight into the pathogenesis of coronary artery stenosis and identify mTOR as a therapeutic target in vasculitis.

**Keywords** Vasculitis; Stenosis; Myofibroblasts; mTOR; Kawasaki Disease
**Subject Categories** Molecular Biology of Disease; Vascular Biology & Angiogenesis

## Introduction

Kawasaki disease (KD) is a medium vessel vasculitis, now recognised as the leading cause of acquired childhood heart disease in developed countries (Burns and Glode, 2004; Dietz et al, 2017). The major clinical complication for children with KD is the development of coronary artery disease. This manifests in the first few weeks of the disease as coronary artery aneurysms, which in most cases resolve (Friedman et al, 2016). However, patients who develop larger coronary aneurysms are at significant risk of subsequent cardiac events (Friedman et al, 2016) caused by the pathological remodelling of the coronary arteries (Tsuda et al, 2005; Patel et al, 2015; Friedman et al, 2016). Indeed, most of these KD patients develop coronary artery stenosis (Tsuda et al, 2005), which impose a significant risk of myocardial infarction due to occlusion or thrombosis (Orenstein et al, 2012; Kuijpers et al, 2003; Friedman et al, 2016).

A similar pathological sequence leading to arterial stenosis can occur in other types of vasculitis, such as Takayasu's arteritis (TAK) and Giant cell arteritis (GCA) (Weyand and Goronzy, 2013; Akiyama et al, 2020; Comarmond et al, 2017; Kumar et al, 2007). Histological analysis of inflamed arteries has revealed that myofibroblasts are the dominant population in the intimal layer of occluded arteries in KD and GCA patients (Orenstein et al, 2012; Kuijpers et al, 2003; Wilson et al, 2004; Checchia et al, 1997; Greigert et al, 2022; Watanabe et al, 2020). Consequently, vasculitis-induced arterial stenosis is attributed to the pathogenic accumulation of myofibroblasts within the intimal layer of inflamed arteries (Orenstein et al, 2012; Dietz et al, 2017).

The myofibroblasts that infiltrate the intima during vasculitis are referred to as luminal myofibroblasts (Orenstein et al, 2012; McCrindle et al, 2017). While their role in driving adverse vascular remodelling is accepted, how luminal myofibroblasts develop remains an open question. It has been reported that the phenotype of luminal myofibroblasts populating inflamed temporal arteries of GCA patients overlaps with their adventitial counterparts (Parreau et al, 2021), leading to the suggestion that these populations may be interrelated. This possibility is consistent with earlier observations that adventitial fibroblasts can migrate into the neointima after experimental balloon injury of the carotid artery (Li et al, 2000; Sartore et al, 2001), and argues that luminal myofibroblasts develop from migratory adventitial fibroblasts (Sartore et al, 2001). However, the expression of alpha-smooth muscle actin

[1]WEHI, Melbourne, VIC 3052, Australia. [2]Department of Forensic Medicine, Monash University, Melbourne, VIC 3006, Australia. [3]Victorian Institute of Forensic Medicine, Melbourne, VIC 3006, Australia. [4]Liver & Intestinal Transplant Unit, Austin Health, Melbourne, VIC 3084, Australia. [5]Department of Surgery, The University of Melbourne, Austin Health, Melbourne, VIC 3084, Australia. [6]Department of Surgery, Austin Health, Melbourne, VIC 3084, Australia. [7]Department of Infectious Diseases, Austin Health, Melbourne, VIC 3084, Australia. [8]DonateLife Victoria, Carlton, VIC 3053, Australia. [9]Department of Intensive Care Medicine, Melbourne Health, Melbourne, VIC 3084, Australia. [10]Department of Microbiology and Immunology, The University of Melbourne, The Peter Doherty Institute for Infection and Immunity, Melbourne, VIC 3052, Australia. [11]North Eastern Public Health Unit, Austin Health, Melbourne, VIC 3084, Australia. [12]Rheumatology Unit, The Royal Melbourne Hospital, Parkville, VIC 3050, Australia. [13]University of Melbourne, Department of Medical Biology, Melbourne, VIC 3052, Australia. ✉E-mail: stock.a@wehi.edu.au; wicks@wehi.edu.au

(α-SMA) by luminal myofibroblasts (Shimizu et al, 2013) has led to the suggestion that intimal fibroblasts develop from smooth muscle cells (SMCs), similar to what is reported during atherosclerosis (Shankman et al, 2015). Endothelial cells and pericytes have also been shown to differentiate into myofibroblasts during experimental models of cardiac and kidney disease (Mederacke et al, 2013; Zeisberg et al, 2007; Li et al, 2009; Li and Bertram, 2010), raising the possibility that luminal myofibroblasts may develop locally, from such intimal resident precursors. Thus, at present, the source of luminal myofibroblasts that emerge in vasculitis remains contentious, and may potentially vary between different diseases, tissues and even blood vessel types.

In this study, we sought to define the origin of luminal myofibroblasts that emerge during vasculitis and explore the signalling pathways that control their development. To do so, we have used complementary lineage tracing systems in a mouse model of KD that re-capitulates coronary artery stenosis in human disease. Our findings reveal that luminal myofibroblasts develop from SMCs and their formation (in multiple forms of systemic vasculitis) is driven by activation of the mechanistic target of rapamycin (mTOR) signalling pathway.

# Results

## Collagen-expressing myofibroblasts populate the coronary artery intima during CAWS-induced vasculitis

To investigate the origin of vasculitis-induced luminal myofibroblasts, we employed a murine model of Kawasaki disease (KD), where cardiac vasculitis is induced by injection of the *Candida albicans* water-soluble complex (CAWS) (Nagi-Miura et al, 2006; Tada et al, 2008). Histological analysis of cardiac sections revealed that CAWS-injected mice (analysed at 4–5 weeks post CAWS injection) developed extensive transmural inflammation of the coronary arteries, accompanied by profound collagen deposition within the adventitial and intimal layers (Fig. EV1A). Moreover, by using confocal microscopy, we found that the endothelial (visualised by CD31 expression) and medial (using autofluorescence to visualise the elastic fibres of the media) layers separate during CAWS-induced vasculitis, due to thickening of the intima (Fig. EV1A,B), similar to what has been found in human KD (Orenstein et al, 2012; Checchia et al, 1997; Friedman et al, 2016).

We next determined if intimal thickening coincided with fibroblast infiltration. To definitively identify fibroblasts, we used the Col1a2$^{CreERT2}$ line, where the CreERT2 fusion gene is expressed under the control of a Collagen-1a2 (Col1a2) enhancer-promoter sequence (Zheng et al, 2002). To report Cre expression, Col1a2$^{CreERT2}$ mice were crossed to the R26-stop-eYFP mice (Srinivas et al, 2001) to create Col1a2$^{CreERT2}$ x R26$^{eYFP}$ mice. In this system, CreERT2 will undergo nuclear translocation upon tamoxifen administration and excise the LoxP flanked stop sequence, resulting in constitutive eYFP expression by Col1a2 expressing cells (Fig. EV1C). We validated this system by flow cytometry, showing that eYFP+ cells emerged within the hearts of tamoxifen-treated Col1a2$^{CreERT2/+}$ x R26$^{eYFP/+}$ mice (but not Cre-negative controls or Col1a2$^{CreERT2/+}$ x R26$^{eYFP/+}$ mice without tamoxifen treatment) which expressed the fibroblast-marker PDGFRα (Fig. EV1D). To determine if collagen-expressing fibroblasts infiltrate the coronary artery intima during CAWS-induced vasculitis, Col1a2$^{CreERT2}$ x R26$^{eYFP}$ mice were injected with CAWS and left for

4–5 weeks to allow intimal hyperplasia to develop. Tamoxifen was then administered to label Col1a2+ expressing fibroblasts for analysis by microscopy (Fig. EV1E). Cardiac sections were stained for eYFP to identify Col1a2+ fibroblasts, CD31 to identify endothelial cells and autofluorescence was used to identify the elastic fibres of the media. We found that the hearts of naive mice contained a small population of Col1a2+ fibroblasts that populated the myocardium and adventitial layer of the coronary arteries but were not found within the intimal layer (Fig. EV1F,G). By comparison, the hearts of CAWS-injected mice contained a substantially expanded Col1a2+ fibroblast population which reproducibly localised to the aortic root and were present within both adventitial and intimal layers of the coronary arteries (Fig. EV1F,G). These findings show that collagen-expressing luminal myofibroblasts infiltrate the coronary artery intima during CAWS-induced vasculitis.

## Luminal myofibroblasts do not develop from epicardial-derived, resident fibroblasts

It has been suggested that luminal myofibroblasts develop from adventitial counterparts (Li et al, 2000; Parreau et al, 2021; Sartore et al, 2001). We therefore used lineage tracing systems to determine if adventitial fibroblasts infiltrate the inflamed intima during CAWS-induced vasculitis. To this end, we utilised the Wt1$^{CreERT2}$ line, which has CreERT2 inserted into the Wilms Tumour 1 (Wt1) locus (Zhou et al, 2008). Wt1 is a transcription factor that is temporally restricted to the epicardium during embryogenesis and drives the epithelial-mesenchymal transition (EMT) of this population (Zhou and Pu, 2012; Moore et al, 1999; von Gise et al, 2011). As such, this system allows the labelling of the epicardium and epicardial-derived cells (EPDCs), which others have shown includes resident cardiac fibroblasts (Moore-Morris et al, 2014). To use this system, we crossed the Wt1$^{CreERT2}$ line to R26$^{eYFP}$ mice and administered tamoxifen at E10.5 of pregnancy (Fig. 1A,B) to achieve the selective labelling of the epicardium and its progeny. We confirmed that an eYFP+ cardiac population develops in E10.5 tamoxifen-treated Wt1$^{CreERT2/+}$xR26$^{eYFP/+}$ mice, but not Cre-negative littermates (Fig. 1C). Phenotypic analysis revealed that the vast majority of the eYFP+ population exhibited a fibroblastic phenotype, expressing high levels of PDGFRα and Podoplanin, while being negative for CD31, CD45 and CD146 (Figs. 1C and EV2A). However, a small population of SMCs (defined as CD45-CD31-CD146+Pdpn+ cells) were eYFP+ (Fig. EV2B), suggesting that the epicardium also gives rise to a minor subset of cardiac SMCs.

We next analysed the distribution of Wt1 + /eYFP+ cardiac cells by confocal microscopy. This revealed that eYFP+ cells are widely distributed throughout the epicardial lining and myocardium of naive Wt1$^{CreERT2}$xR26$^{eYFP}$ mice (Fig. 1E). Notably, Wt1 + /eYFP+ cells were enriched around the proximal coronary arteries and populated the adventitial layer, confirming robust labelling of adventitial fibroblasts using this system (Fig. 1E). To determine if these adventitial fibroblasts infiltrate the intima during vasculitis, adult Wt1$^{CreERT2}$ x R26$^{EYFP}$ mice (labelled at E10.5) were injected with CAWS and their hearts analysed by microscopy 4–5 weeks later (Fig. 1D). This analysis revealed a significant increase in eYFP+ cells within the adventitial layer around the coronary arteries of CAWS-injected mice (Fig. 1E,F). However, the intimal layer of the coronary artery was devoid of Wt1 + /eYFP+ EPDCs (Fig. 1E,F). These findings show that while epicardial-derived resident fibroblasts expand within the adventitia of the coronary arteries during vasculitis, this population does not migrate into the

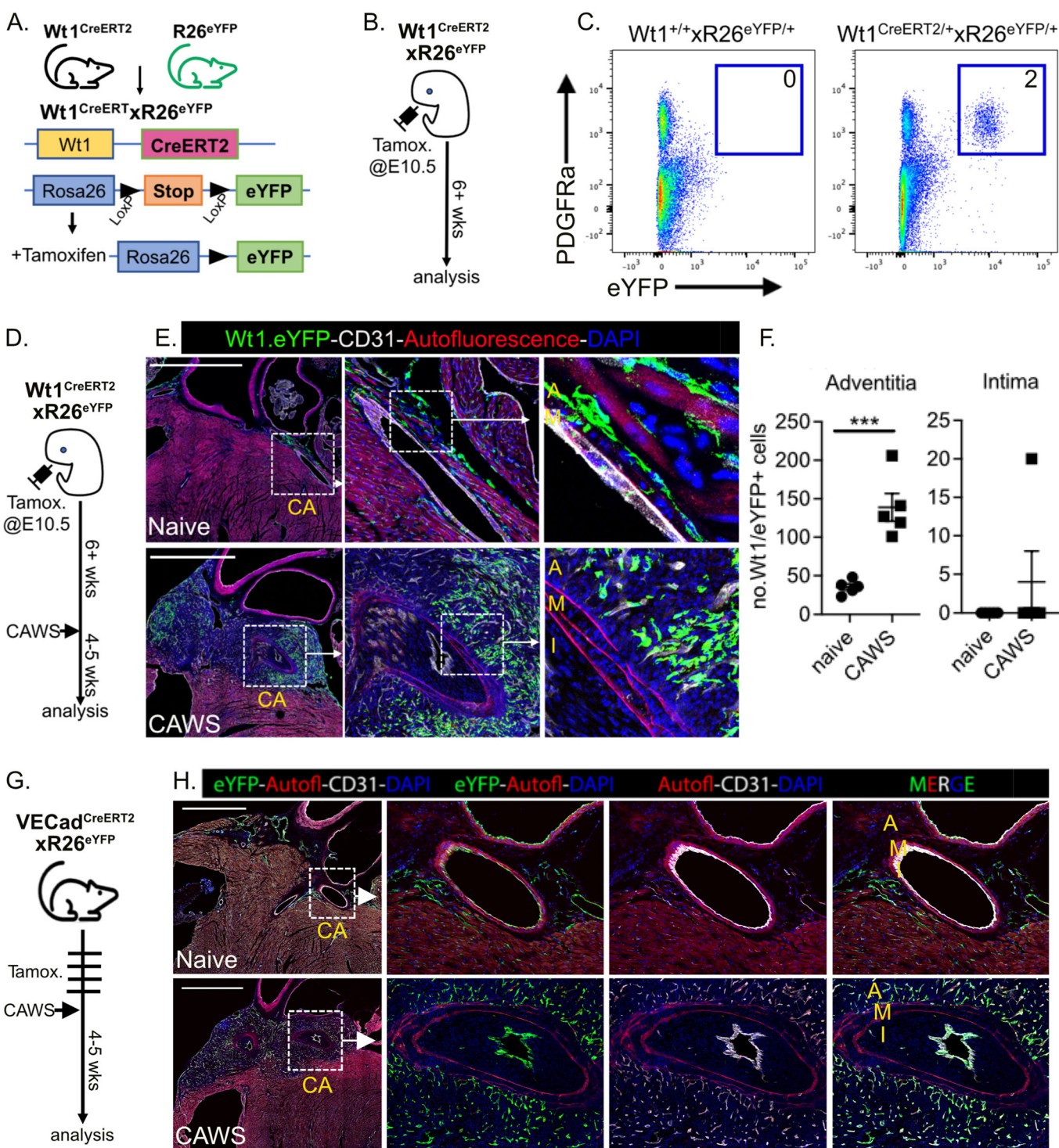

intima. Thus, the luminal myofibroblasts that emerge in the CAWS model of KD develop independently from adventitial fibroblasts.

## Luminal myofibroblasts develop independently from endothelial cells

Endothelial cells have also been reported to give rise to myofibroblasts through endothelial-to-mesenchymal transition (endo-MT) (Li et al, 2009; Zeisberg et al, 2007). We therefore investigated if endothelial cells form luminal myofibroblasts. To trace endothelial cells, we crossed VECad^CreERT2 mice (where CreERT2 is driven by the vascular endothelial cadherin promoter) to the R26^eYFP reporter line, to achieve tamoxifen-inducible eYFP labelling of endothelial cells. To determine if this population undergoes endo-MT during vasculitis, tamoxifen-treated VECad^CreERT2xR26^eYFP mice were injected with CAWS and their

◄ **Figure 1. Luminal myofibroblasts develop independently of both epicardial-derived, adventitial fibroblasts and endothelial cells.**

(A, B) Genetic and experimental schema for Wt1[CreERT2].R26[eYFP] system. (C) Flow cytometric analysis of cardiac cells from Wt1[CreERT2].R26[eYFP] (and control) mice showing endogenous eYFP+ expression and anti-PDGFRα staining (mean of 5 mice/group from three independent experiments is inset). (D) Experimental schema. (E) Cardiac sections from naïve or CAWS-injected Wt1[CreERT2].R26[eYFP] mice (4–5 weeks post injection) analysed by immunofluorescent microscopy. Sections were stained for GFP to identify Wt1+/eYFP+ cells (green), CD31 to label endothelial cells (white) and autofluorescence to identify elastin fibres of the media (red). (F) Graphs show the number of Wt1+/eYFP+ cells within the coronary artery adventitia and intima for individual mice (with mean ± SEM) pooled from three independent experiments. (G) Experimental schema for VECad[CreERT2].R26[eYFP] system. (H) Representative cardiac sections (of 5–6 mice per group, three independent experiments) from naïve or CAWS-injected VECad[CreERT2].R26[eYFP] mice analysed by immunofluorescent microscopy as above. Throughout, the coronary artery (CA), adventitia (A), media (M) and intima (I) are annotated and scale bars are 1000 μm. *** $P < 0.001$ with two-tailed Student's $t$ tests. Exact $P$ value (to 4 decimal points) for (F) 0.0005 (***). Source data are available online for this figure.

hearts analysed by confocal microscopy 4–5 weeks later (Fig. 1G). Despite profound intimal hyperplasia of the coronary arteries in CAWS-injected mice, VECad+/eYFP+ cells remained localised to the luminal lining of the coronary artery, colocalising with the CD31+ endothelial cells (Fig. 1H). Thus, endothelial cells did not infiltrate the intima during vasculitis, illustrating that luminal myofibroblasts do not develop through endo-MT.

## Luminal myofibroblasts develop from Myh11 + SMC precursors

Smooth muscle cells (SMCs) have been shown to form intimal fibroblasts during atherosclerosis (Shankman et al, 2015), prompting us to examine whether a similar process occurs during vasculitis. To trace SMCs, we used Myh11[CreERT2] mice (Wirth et al, 2008), where the inducible CreERT2 is expressed under the control of the enhancer-promoter region from the smooth muscle myosin polypeptide 11 (*Myh11*), a contractile protein highly expressed by SMCs. To achieve tamoxifen-inducible eYFP labelling of Myh11+ SMCs, Myh11[CreERT2] mice were crossed to the R26[eYFP] line (Fig. 2A). Flow cytometric analysis confirmed that eYFP+ cells emerged within the hearts of tamoxifen-treated, Myh11[CreERT2]xR26[eYFP] mice but not Cre-negative controls (Fig. 2B,C). Phenotypic analysis revealed that the Myh11+/eYFP+ cardiac population was negative for phenotypic markers of endothelial cells (CD31), leukocytes (CD45) and fibroblasts (Pdpn, PDGFRα) but was uniformly positive for the mural cell marker CD146 (Fig. 2D), confirming that Myh11[CreERT2] recombination labels cardiac SMCs.

To determine if SMCs form luminal myofibroblasts during vasculitis, Myh11[CreERT2] x R26[eYFP] mice were administered tamoxifen to label SMCs, injected with CAWS and hearts analysed 4–5 weeks later by confocal microscopy (Fig. 2E). As expected, Myh11+/eYFP+ cells were found exclusively within the medial layer of the coronary artery arteries of naïve mice (Fig. 2F). In contrast, Myh11-derived eYFP+ cells extensively populated the inflamed coronary artery intima of CAWS-injected mice, forming a dominant population within the thickened intima (Fig. 2F,G). In addition, transcriptional analysis by RT-PCR, revealed that eYFP+ SMC-derived cells sorted from the hearts of CAWS-injected mice had increased expression of fibrillar collagens (*Col1a1, 1a2, Col3a1*), enzymes involved in collagen synthesis (P4HA3) and inflammatory cytokines (SAA3, OPN) and chemokines (Ccl2/7; Fig. 2H). Collectively, these findings demonstrate that during vasculitis, SMCs acquire the ability to undergo media-to-intimal migration and drive fibrosis and inflammation. These properties culminate in the formation of pathogenic luminal myofibroblasts which drive vascular remodelling.

## Phenotypic analysis of luminal myofibroblasts

We next performed co-labelling studies to investigate the phenotype of luminal myofibroblasts. Analysis of cardiac sections from CAWS-injected Myh11[CreERT2] x R26[eYFP] mice revealed that eYFP+ luminal myofibroblasts, and particularly those located near the lumen, expressed α-SMA and CD146, and had minimal PDGFRα expression (Fig. EV3A). We next investigated the phenotype of luminal myofibroblasts in humans by analysing the autopsy tissue of two infants who died from myocardial infarction during acute KD. We found that in both KD cases, the inflamed coronary artery intima was heavily populated by α-SMA+ myofibroblasts that were uniformly CD146 positive (Fig. EV3B). Notably, α-SMA+ myofibroblasts of the adventitia were CD146 negative, exhibiting a distinct phenotype from their luminal counterparts (Fig. EV3B). These findings provide further evidence (in humans) that luminal myofibroblasts develop independently of adventitial myofibroblasts, and instead have a SMC origin.

## The formation of SMC-derived luminal myofibroblasts coincides with activation of the mTOR signalling pathway

We and others have reported that the mechanistic target of the rapamycin (mTOR) pathway becomes activated by multiple populations during vasculitis (Stock et al, 2023; Hadjadj et al, 2018; Maciejewski-Duval et al, 2018; Wen et al, 2017). This prompted us to examine whether luminal myofibroblasts utilise this signalling pathway. To address this question, cardiac sections from Myh11[CreERT2] x R26[eYFP] mice were stained for phosphorylated ribosomal protein S6 (pS6), a downstream target of (and biomarker for) active mTOR signalling (Liu and Sabatini, 2020). We analysed cardiac sections from naïve controls and CAWS mice 21 days after injection, a timepoint when luminal myofibroblasts are first emerging (Fig. 3A). By staining for pS6, eYFP and Ki67 (to identify proliferating cells), we found that in naïve mice, Myh11+/eYFP+ cells were restricted to the media and negative for pS6 and Ki67, indicating minimal mTOR signalling or proliferation in quiescent SMCs (Fig. 3B,C). By comparison, at day 21 post CAWS injection, Myh11+/eYFP+ cells had begun to infiltrate the intima and activate mTOR signalling, as shown by strong pS6 staining. Notably, while pS6 staining was most striking in eYFP+ cells located adjacent to the endothelial layer (Fig. 3B), the majority of eYFP+ cells in both the media and intima were pS6+ of CAWS mice (Fig. 3C). Thus, Myh11-derived cells in both layers of the vessel activate mTOR signalling during vasculitis. Finally, we found that some pS6+eYFP+ cells were Ki67+, linking mTOR activation with SMC proliferation.

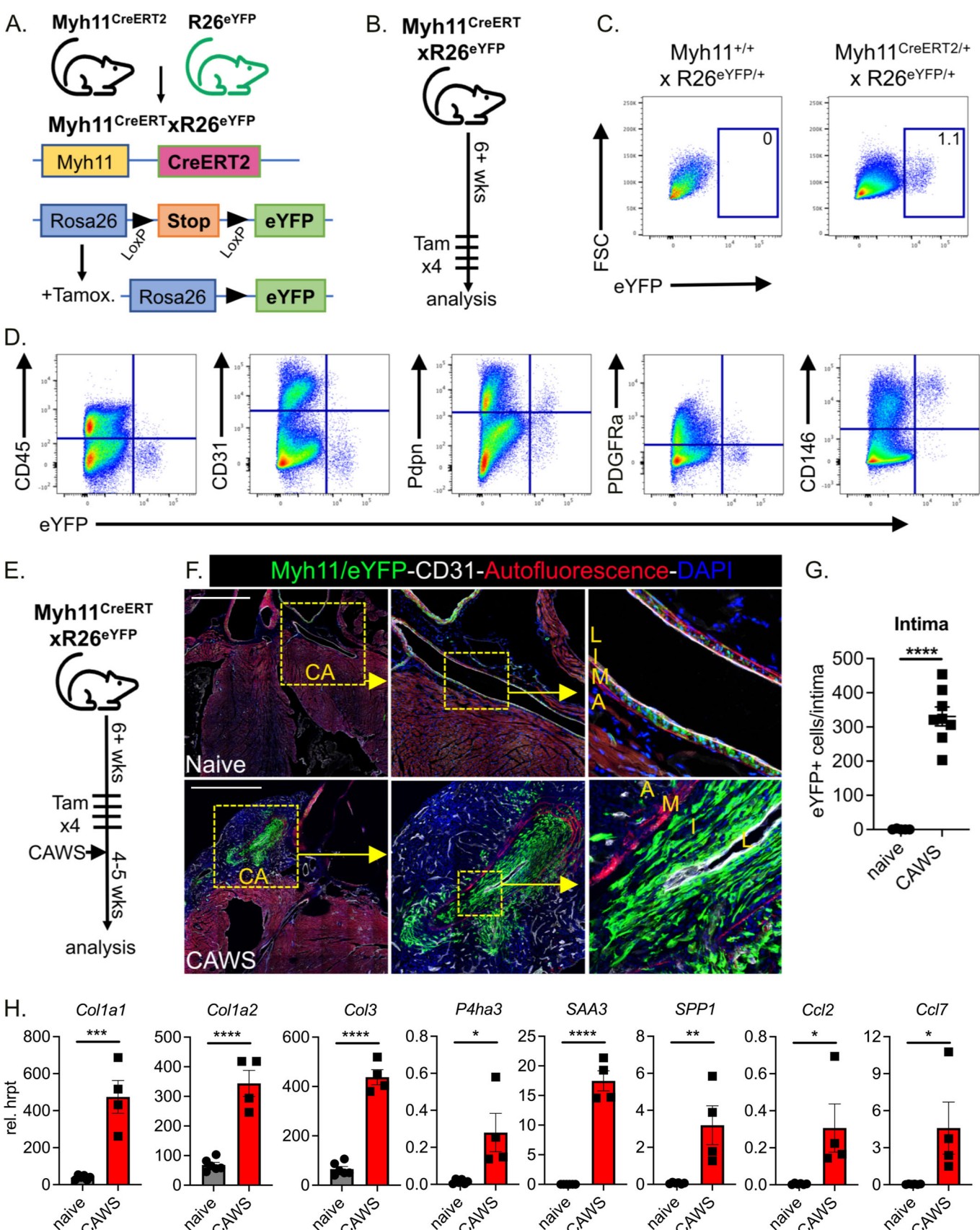

**Figure 2.  Luminal myofibroblasts develop from Myh11 + SMCs during CAWS-induced vasculitis.**

(A, B) Genetic and experimental schema for the Myh11^CreERT2^.R26^eYFP^ system. (C, D) Flow cytometric analysis of cardiac cells from Myh11^CreERT2^.R26^eYFP^ (and control) mice showing endogenous eYFP+ expression (with the mean of 5–8 mice, three independent experiments inset; (C)) with lineage-specific markers (D). (E) Experimental schema. (F) Cardiac sections from naive or CAWS-injected Myh11^CreERT2^.R26^eYFP^ mice analysed by immunofluorescent microscopy. Sections were stained for GFP to identify Myh11 + /eYFP+ cells (green), CD31 to label endothelial cells (white) and autofluorescence to identify elastin fibres of the media (red). (G) Graphs show the number of Myh11 + /eYFP+ cells within the coronary artery intima for individual mice (with mean ± SEM) pooled from three independent experiments. (H) RT-PCR analysis of CD146 + /eYFP+ cells isolated from the hearts of naive and CAWS-injected Myh11^CreERT2^.R26^eYFP^ mice. Data shows the mean ± SEM from 2 to 3 pools of mice (2–4 mice/pool) acquired in three independent experiments. Throughout, the coronary artery (CA), adventitia (A), media (M), intima (I) and lumen (L) are annotated and scale bars are 1000 μm. *$P$ < 0.05; **$P$ < 0.01; ***$P$ < 0.001; ****$P$ < 0.0001 with two-tailed Student's $t$ tests. Exact P values (to 4 decimal points) for: (G) < 0.0001 (****); (H) Col1a1 0.0003 (***), Col1a2 < 0.0001 (****), Col3 < 0.0001 (****), P4ha3 0.0119 (*), SAA3 < 0.0001 (****), SPP1 0.0054 (**), Ccl2 0.0188 (*) and Ccl7 0.0258 (*). Source data are available online for this figure.

## IL-1β activates mTOR signalling and SMC-dependent effector functions

We explored what upstream signals may drive mTOR activation in SMCs. We tested IL-1β, IL-17, Tnf and GM-CSF, which are cytokines known to have critical roles in driving cardiac vasculitis in KD (Lee et al, 2015; Lin et al, 2024; Stock et al, 2016; Stock et al, 2019). For these experiments, we treated murine aortic SMCs with cytokines (with or without the mTOR inhibitor rapamycin) and stained for phospho-S6 and SAA3, a cytokine we found to be highly expressed by activated SMCs during CAWS-induced vasculitis (Fig. 2H). We found that both IL-1β and Tnf (but not IL-17 or GM-CSF) triggered mTOR signalling, as revealed by increased pS6 (Fig. EV4B). Moreover, we found that IL-1β (but not Tnf) induced SAA3 expression by SMCs, which was reduced by rapamycin (Fig. EV4C). These findings show that IL-1β can directly activate mTOR signalling in SMCs and drive effector functions (i.e., SAA3 production). Consistent with an earlier study (Porritt et al, 2021), these findings raise the possibility that an IL-1-mTOR axis promotes SMC activation.

## Luminal myofibroblasts activate mTOR signalling in patients with Kawasaki disease, giant cell arteritis and Takayasu's arteritis

In view of these findings in a murine model of vasculitis, we were keen to examine if the mTOR pathway is also activated in the luminal myofibroblasts that emerge in human vasculitis. We first analysed coronary artery sections from autopsy tissue of two KD fatalities (also shown in Fig. EV3B). H&E staining revealed that in comparison to controls obtained from cardiac-disease-free organ donors, the coronary arteries from both KD cases showed extensive immune cell infiltrate, which coincided with regions of intimal hyperplasia (Fig. 4A). To measure mTOR signalling by luminal myofibroblasts, coronary artery sections were stained for α-SMA (to identify myofibroblasts) and pS6 (to measure mTOR activity) and analysed by confocal microscopy. Control arteries from organ donors showed minimal pS6 staining in α-SMA+ cells within the media and intima, indicating limited mTOR signalling in the steady state (Fig. 4B). In comparison, both KD cases showed strong regional expression of pS6+ by α-SMA+ fibroblasts within the inflamed intima (Fig. 4B). Thus, mTOR signalling pathway is activated by the luminal myofibroblasts that populate the coronary arteries in acute KD.

We next investigated whether the luminal myofibroblasts that emerge in other forms of vasculitis also activate the mTOR signalling pathway. We first examined Takayasu's arteritis (TAK), a large vessel vasculitis involving the aorta and its major branches (Weyand and Goronzy, 2013). H&E analysis of coronary artery sections from two TAK fatalities revealed profound remodelling of the coronary arteries, associated with intimal hyperplasia and immune cell infiltrate (Fig. 5A). Probing for mTOR signalling by confocal microscopy revealed that pS6 + /α-SMA+ intimal myofibroblasts were evident in one of the TAK cases (TAK case 1; Fig. 5B). We postulate that this region of pS6+ myofibroblasts corresponds to an area of nascent vascular remodelling, and that mTOR signalling may also be a feature of intimal fibroblast activation in TAK.

We also examined sections from patients who underwent temporal artery biopsy for possible giant cell arteritis (GCA). GCA is a common form of systemic vasculitis that occurs in the elderly and can affect the aorta and its major branches (Weyand and Goronzy, 2013; Akiyama et al, 2020). We analysed two pathologist-confirmed GCA positive cases together with a GCA negative, control temporal artery (Fig. 5C). Confocal analysis revealed that while α-SMA+ cells within the normal temporal artery were pS6 negative, a large proportion of α-SMA+ luminal myofibroblasts within the inflamed temporal arteries of both GCA cases showed strong pS6 staining (Fig. 5D). Collectively, these findings illustrate that the mTOR signalling is activated by the luminal myofibroblasts that drive adverse vascular remodelling in patients with KD, TAK and GCA.

## mTORC1 signalling is an intrinsic and essential regulator of SMC-derived luminal myofibroblast formation that can be targeted through pharmacological inhibition

To address whether mTOR signalling directly controls the development of luminal myofibroblasts during vasculitis, we asked whether removing mTOR signalling from SMCs abrogated their development? To disrupt mTOR signalling, we genetically targeted the Regulatory Associated Protein of mTOR (raptor), which is a major subunit of the mTORC1 complex (Nojima et al, 2003; Liu and Sabatini, 2020; Bentzinger et al, 2008). To this end, we utilised raptor-floxed mice, which have exon 6 of the raptor locus flanked by LoxP sites, allowing Cre-mediated *raptor* deletion (Bentzinger et al, 2008). To drive SMC-specific *raptor* deletion, raptor^fl/fl^ mice were crossed to the Myh11^CreERT2^ line. Notably, the *raptor*^fl/fl^ line were also bred to carry the R26-stop-eYFP, meaning that Myh11-driven recombination will simultaneously delete *raptor* and activate eYFP transcription, allowing the identification of *raptor*-deficient SMCs through eYFP expression (Fig. 6A). To investigate

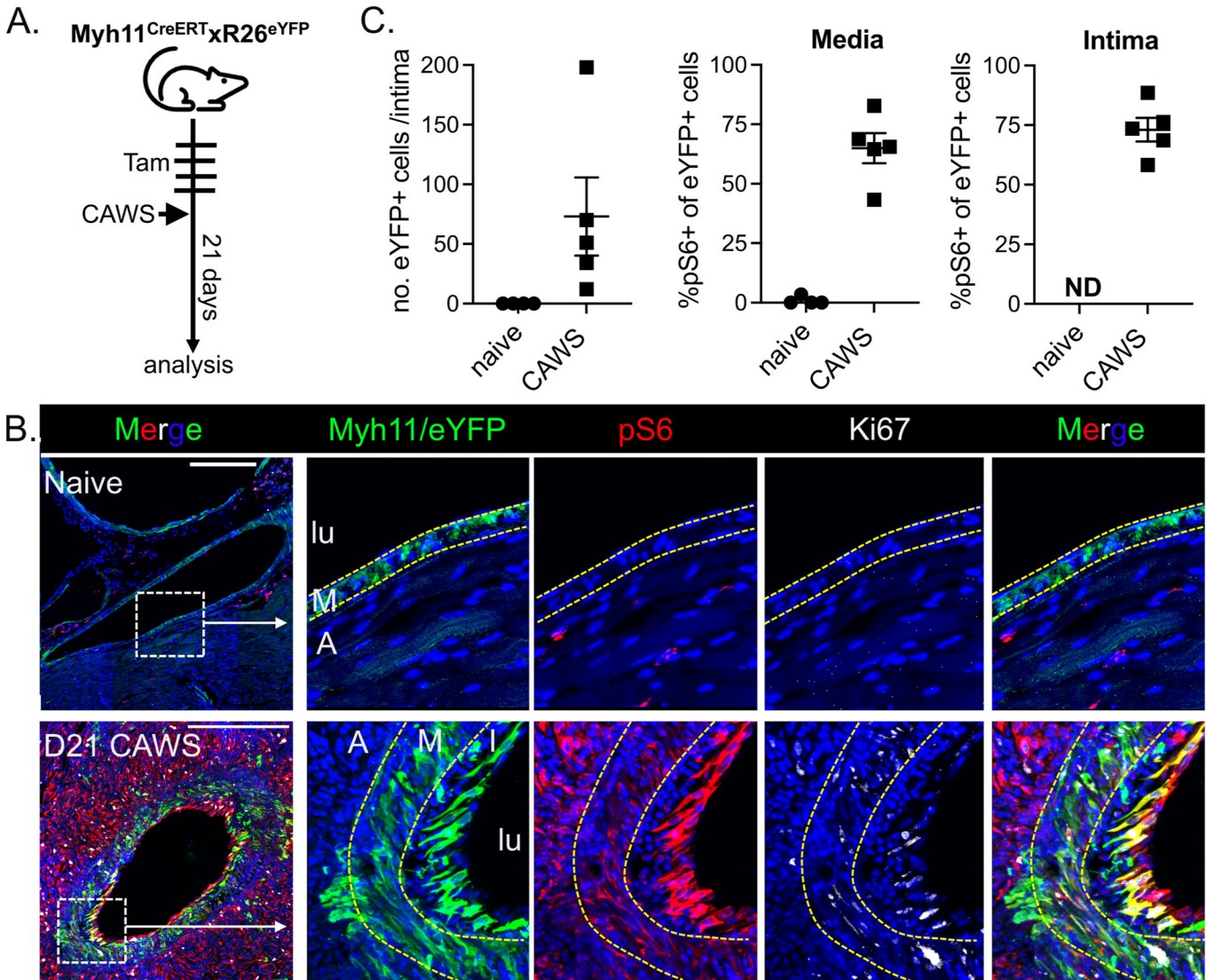

**Figure 3. Myh11 + SMCs activate the mTOR signalling pathway during vasculitis.**

(A) Experimental schema. (B) Cardiac sections from naive or day 21 CAWS-injected Myh11$^{CreERT2}$.R26$^{eYFP}$ mice analysed by immunofluorescent microscopy. Sections were stained for GFP to identify Myh11 + /eYFP + cells (green), pS6 (red) and Ki67 (white). (C) Graphs show the number of Myh11 + /eYFP + cells within the coronary artery intima or % of Myh11 + /eYFP + cells that are pS6 + within the media and intimal layers of the coronary artery. Dots represent individual mice (with mean ± SEM) pooled from two independent experiments. The coronary artery (CA), adventitia (A), media (M), intima (I) and lumen (L) are annotated and scale bars are 250 μm. Source data are available online for this figure.

whether SMC-specific Raptor deletion impacts luminal myofibroblast formation, Myh11$^{CreERT2}$xRaptor$^{fl/fl}$xR26$^{eYFP}$ mice (where raptor is deleted from eYFP + /SMCs) and Myh11$^{CreERT2}$xRaptor$^{+/+}$ xR26$^{eYFP}$ controls (where raptor is intact) were administered tamoxifen to trigger recombination in SMC, and then injected with CAWs to induce vasculitis (Fig. 6B). After 4–5 weeks, hearts were analysed by histology and confocal microscopy. H&E staining revealed that CAWS injection induced comparable levels of cardiac inflammation in both Raptor$^{+/+}$ and Raptor$^{fl/fl}$ groups, as measured by the cardiac area with immune cell infiltrate (Fig. 6C,F). As in previous experiments, confocal microscopy showed that in control Myh11$^{CreERT2}$ x Raptor$^{+/+}$xR26$^{eYFP}$ mice (where raptor was intact), Myh11 + /eYFP + cells robustly infiltrated the coronary artery

intima following CAWS injection, causing profound arterial stenosis (Fig. 6D,E). In stark contrast, the Myh11 + /eYFP + cells of Myh11$^{CreERT2}$ x Raptor$^{fl/fl}$xR26$^{eYFP}$ mice (where raptor was deleted in SMCs) were entirely absent from the coronary artery intima and remained within the medial layer (Fig. 6D,E). In control experiments, we confirmed that raptor-deficient eYFP + SMCs were pS6 negative, illustrating that mTOR signalling is silenced in these cells (Fig. EV5A,B). Thus, deleting raptor from SMCs completely abrogated their ability to migrate into the coronary artery intima during vasculitis. Moreover, we found that SMC-raptor-deficient mice had significantly reduced arterial stenosis following CAWS injection compared to Raptor + /+ controls (Fig. 6G). Indeed, the residual stenosis that does develop in SMC-

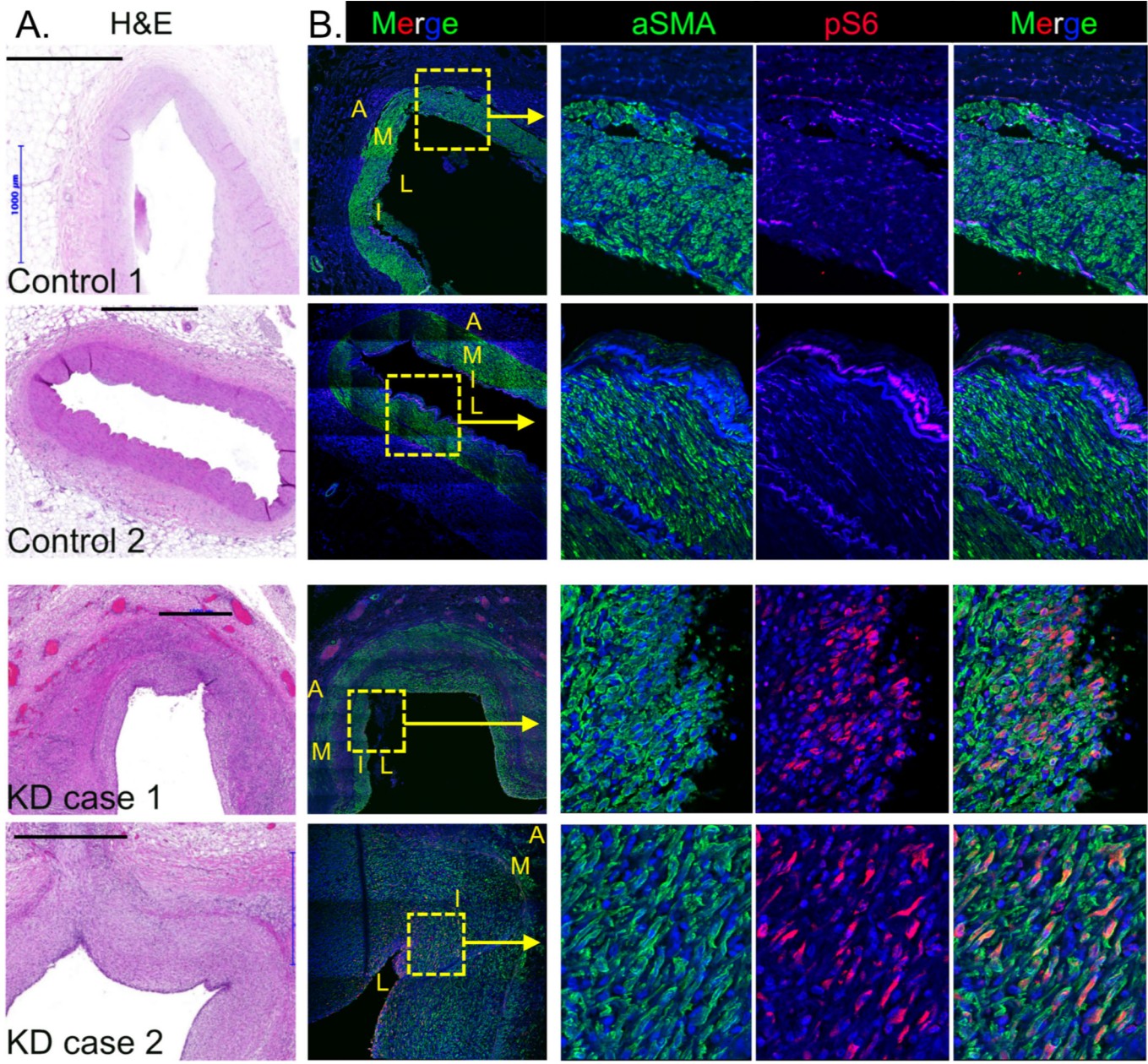

**Figure 4. Luminal myofibroblasts of KD patients have an activated mTOR signalling pathway.**

Coronary artery sections from two cardiac-disease-free organ donors or two acute KD fatalities analysed by H&E staining (**A**) or immunofluorescent microscopy (**B**). For immunofluorescence, sections were stained for α-SMA (green) and pS6 (red) and analysed by confocal microscopy. Box shows inset area and the adventitia (A), media (M), intima (I) and lumen (L) are annotated and scale bars are 1000 μm. Source data are available online for this figure.

raptor-deficient mice is driven by the infiltration of eYFP negative α-SMA+ cells (Fig. EV5C). The lack of eYFP expression indicates that these are α-SMA+ myofibroblasts where recombination (and hence raptor deletion) has not occurred, meaning mTOR is intact. Collectively, these findings illustrate that mTORC1 signalling is an intrinsic and essential process for the development of SMC-derived luminal myofibroblasts and that attenuating luminal myofibroblast formation (via targeting mTOR) reduces arterial stenosis in vasculitis.

Finally, we explored the therapeutic potential of targeting mTORC1. To this end, we examined whether the pharmacological inhibition of mTOR signalling (with rapamycin) can block the formation of SMC-derived luminal myofibroblasts. To this end, tamoxifen-treated Myh11[CreERT2] x R26[eYFP] mice were injected with CAWS to induce vasculitis and 8 days later, were treated continuously (3x/week) with either the mTOR inhibitor Rapamycin or vehicle control, until day 28 post CAWS (Fig. 7A). The analysis of cardiac sections revealed that eYFP+ SMC-derived cells had

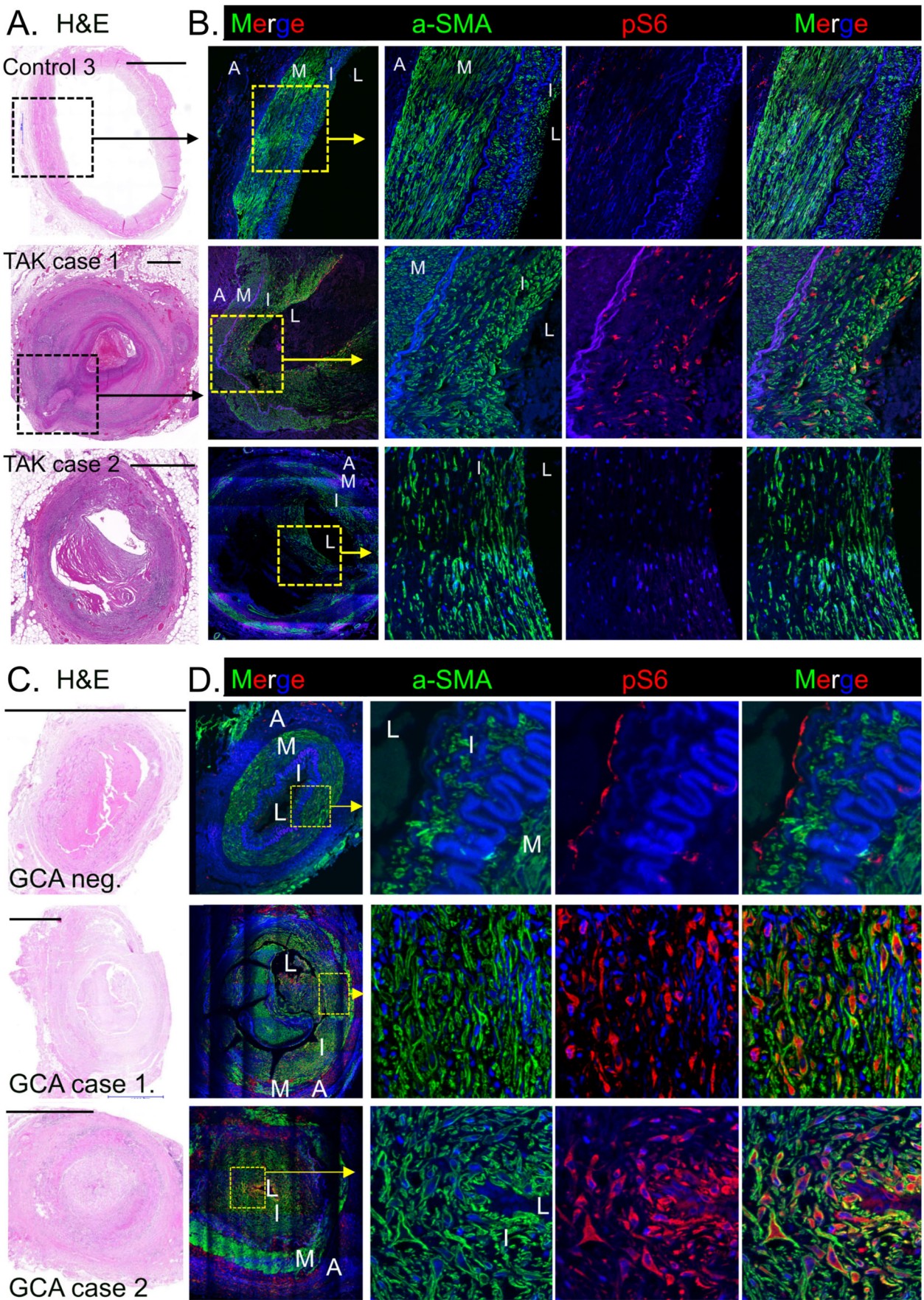

◄  **Figure 5.  Luminal myofibroblasts of patients with Takayasu's arteritis and giant cell arteritis have an activated mTOR signalling pathway.**

(A, B) Coronary artery sections from one cardiac-disease-free organ donor (control 3) and two Takayasu's arteritis fatalities analysed by H&E staining (A) or immunofluorescent microscopy (B). For immunofluorescence, sections were stained for α-SMA (green) and pS6 (red) and analysed by confocal microscopy. (C, D) Temporal artery sections from a GCA negative control and two GCA -positive patients analysed by H&E staining (C) or immunofluorescent microscopy (D) as above. Box shows inset area and the adventitia (A), media (M), intima (I) and lumen (L) are annotated and scale bars are 1000 μm. Source data are available online for this figure.

richly migrated into the inflamed coronary artery intima of vehicle-treated CAWS mice, causing profound arterial stenosis (Fig. 7B–D). In comparison, eYFP+ SMCs remained within the coronary artery media of rapamycin-treated mice, coinciding with minimal stenosis (Fig. 7B–D). These findings demonstrate that pharmacological mTOR inhibition can block the activation and/or migration of SMC-derived myofibroblasts during vasculitis, and in doing so, prevent pathological vascular remodelling.

## Discussion

We report two major findings into the luminal myofibroblasts that drive vascular remodelling in vasculitis. Using multiple lineage tracing systems in a mouse model of KD, we provide definitive evidence that vasculitis-induced luminal myofibroblasts develop from migratory SMCs. Moreover, we exclude both epicardial-derived resident fibroblasts and endothelial cells as potential sources, illustrating that neither adventitial-to-intimal migration (by fibroblasts) nor endothelial-to-mesenchymal transition (endo-MT) occurs during vasculitis. However, a potential caveat of our study is that the Wt1$^{CreERT2}$ system does not label all resident fibroblasts, due to either suboptimal recombination (in Wt1+ epicardial cells) or the fact that fibroblasts can also develop from alternate sources, such as the endocardium (Moore-Morris et al, 2014). Thus, it is possible that adventitial fibroblasts not labelled in Wt1$^{CreERT2}$ system may form luminal myofibroblasts. Nevertheless, despite this limitation, our findings with the Myh11$^{CreERT2}$ system clearly show that a majority of luminal myofibroblasts emerge from SMCs that transform into a pathogenic phenotype during vasculitis, characterised by the ability to migrate, proliferate and produce fibrotic and inflammatory factors.

Our findings are consistent with earlier studies in the *Lactobacillus Casei* cell wall extract (LCWE) model of KD showing that SMCs undergo a phenotypic switch during vasculitis associated with the upregulation of fibrotic and proliferative markers and the increased expression of MMPs and inflammatory cytokines/chemokines (Zhang et al, 2020; Porritt et al, 2021). Critically, these earlier studies demonstrate that inhibiting SMC activation (or dedifferentiation) reduced the severity of cardiac vasculitis and medial thickening (Porritt et al, 2021; Zhang et al, 2020). Our findings that inhibiting SMC activation significantly attenuated coronary artery stenosis are consistent with these earlier studies (Zhang et al, 2020; Porritt et al, 2021), which together, provide proof-of-principle that limiting SMC activation can improve clinical outcomes in KD.

Thus, SMC play a critical role in driving vascular pathology in vasculitis. Intriguingly, SMCs can develop from multiple origins during embryogenesis (Wang et al, 2015). While those of the coronary artery are typically described as developing from epicardial precursors (Wang et al, 2015; Dettman et al, 1998), it

is important to note that SMCs of the proximal coronary arteries—where vasculitis develops in KD – have been reported to instead emerge from neural crest cells (Arima et al, 2012; Jiang et al, 2000). Indeed, our findings that relatively few SMCs of the proximal coronary arteries are labelled in Wt1$^{CreERT2}$ system, supports a non-epicardial origin. Thus, while further studies are required with neural crest-specific lineage tracing, we predict that the luminal myofibroblasts that emerge during KD most likely develop from neural crest-derived SMCs. In this regard, it is intriguing to note that neural crest-derived SMCs have enhanced collagen and proliferative responses to cytokines (compared to their mesoderm-derived counterparts) (Topouzis and Majesky, 1996; Gadson et al, 1997), raising the possibility that the susceptibility of the proximal coronary arteries to KD may be (at least partly) attributed to the unique potential of neural crest-derived SMCs.

Our second major observation is to show that the formation of SMC-derived luminal myofibroblasts is driven by activation of the mTOR signalling pathway. mTOR is a protein kinase that can form two distinct multimeric signalling complexes, mTORC1 and mTORC2, each composed of mTOR with various binding partners (Liu and Sabatini, 2020). The mTORC1 complex includes raptor and promotes cell growth and proliferation by amplifying protein translation and lipid biogenesis through the phosphorylation of downstream targets, such as S6K1 (p70-S6 Kinase 1) and 4E-BP1 (the eukaryotic initiation factor 4E binding protein 1) (Liu and Sabatini, 2020). Here, we show that the mTOR signalling cascade is activated by luminal myofibroblasts that emerge in patients with Kawasaki disease, Takayasu's arteritis and giant cell arteritis. Moreover, SMCs where mTORC1 complex was selectively deleted (through the deletion of raptor) completely failed to populate the inflamed intima during vasculitis. These findings demonstrate that signalling through the mTORC1 complex is: (i) activated in different types of human vasculitis and (ii) plays an intrinsic and essential role in driving the development of pathogenic luminal myofibroblasts.

Our findings raise mechanistic questions and present clinical opportunities. Foremost is how does mTOR control the formation of luminal myofibroblasts? Our finding that raptor-deficient SMCs remain within the media, despite vascular inflammation, indicates that mTOR signalling most likely controls the migration potential of SMCs. Indeed, previous studies have reported that rapamycin inhibits SMC migration in vitro (Poon et al, 1996), and our findings corroborate this in vivo. However, the molecular basis for how SMCs undergo media-to-intimal migration, and how this process is regulated by mTOR requires further investigation. In addition, it is noteworthy that mTOR signalling persists after intimal invasion. This finding implies that mTOR may also regulate other pathological features of luminal myofibroblasts beyond SMC migration, such as cellular proliferation, effector functions (such as fibrosis or inflammation) and/or survival. Thus, while further mechanistic studies are required, we predict that mTOR may play a

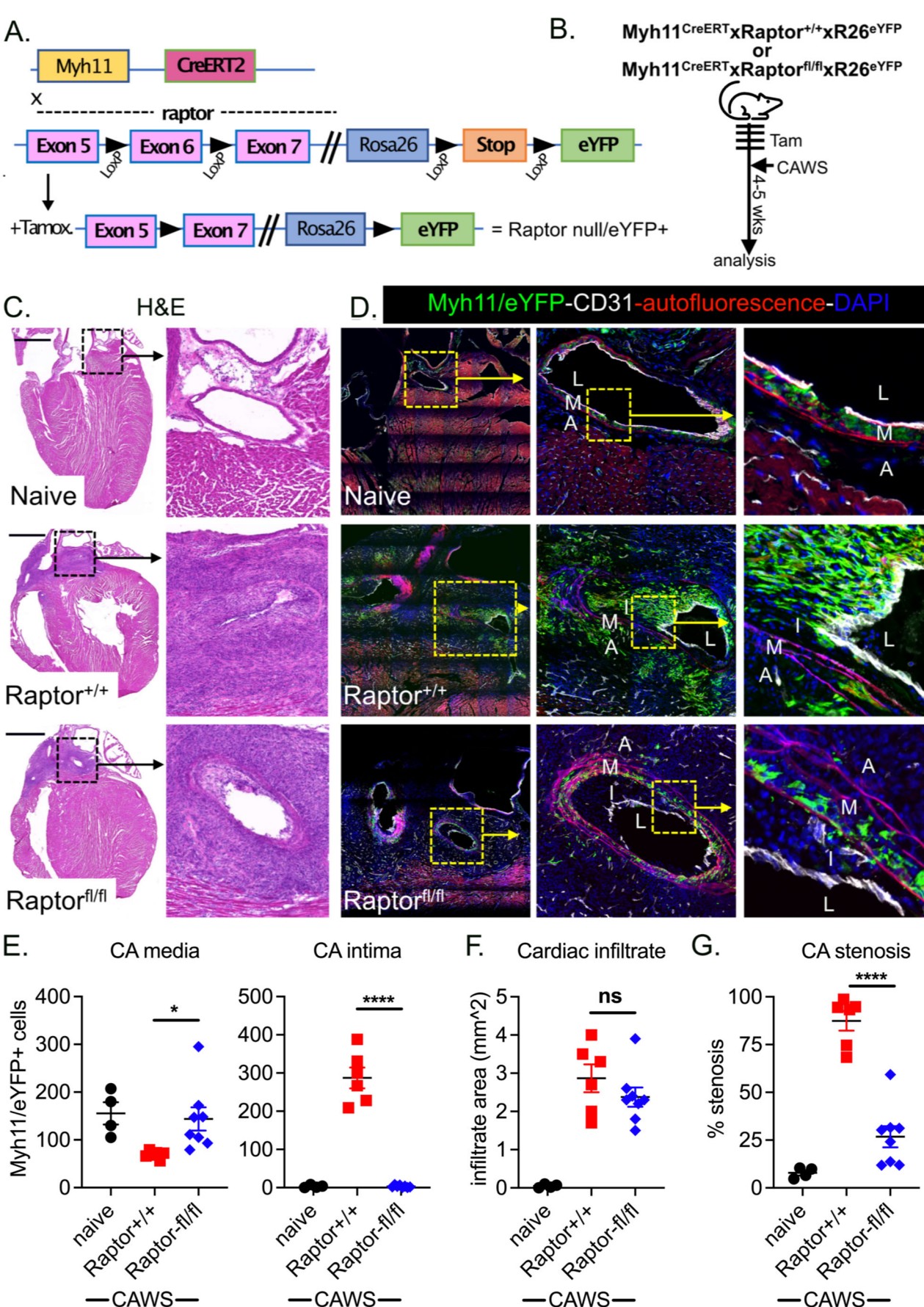

◀ **Figure 6. mTORC1 is an intrinsic and essential regulator of SMC-derived luminal myofibroblast formation.**

(A, B) Genetic and experimental schema for Myh11^CreERT2.Raptor^fl/fl.R26^eYFP system. (C, D) Cardiac sections from naive or CAWS-injected Myh11^CreERT2.Raptor^fl/fl.R26^eYFP or Myh11^CreERT2.Raptor^+/+.R26^eYFP mice (4–5 weeks post injection) analysed by H&E staining (C) or immunofluorescent microscopy (D). For immunofluorescence, sections were stained for GFP to identify Myh11 + /eYFP+ cells (green), CD31 to label endothelial cells (white) and autofluorescence to identify elastin fibres of the media (red). (E–G) Graphs show the number of Myh11 + /eYFP+ cells within the coronary artery media or intima (E), the total cardiac area with H&E+ immune cell infiltrate (F) or CA stenosis (G). Points depict individual mice (with mean ± SEM) pooled from three independent experiments. The coronary artery (CA), adventitia (A), media (M), intima (I) and lumen (L) are annotated and scale bars are 1000 μm. *$P < 0.05$; ****$P < 0.0001$ with two-tailed Student's *t* tests (ns denotes non-significant difference). Exact *P* values (to 4 decimal points) for: (E) media 0.0222 (*), intima <0.0001 (****), (F) 0.2728 (ns), (G) < 0.0001 (****). Source data are available online for this figure.

multifaceted role in controlling the development and pathological effector functions of luminal myofibroblasts.

Our findings also raise the question of which factors trigger mTOR activation during vasculitis? IL-1 has recently been reported to drive SMC activation in the *Lactobacillus Casei* cell wall extract (LCWE) mouse model of KD (Porritt et al, 2021). Notably, we show that in vitro, IL-1β can directly activate mTOR signalling in SMCs and drive effector functions (i.e., SAA3 production). Thus, together with this earlier study (Porritt et al, 2021), these findings raise the possibility that an IL-1-mTOR axis promotes the luminal myofibroblasts formation. In addition, Notch signalling has also been reported to drive mTOR activation by fibroblasts cells in rheumatoid arthritis (Wei et al, 2020), and T cells in vasculitis (Wen et al, 2017). Intriguingly, in these instances, the Notch activating ligand Jagged-1, was reported to be expressed on activated endothelial cells (Wen et al, 2017; Wei et al, 2020). Given our observations that luminal myofibroblasts adjacent to endothelial cells show high mTOR activity (as seen by increased pS6 staining), it is possible that activated endothelial cells may drive myofibroblast activation through a Notch-mTOR signalling axis. Identifying the upstream regulators of mTOR could identify novel therapeutic options to limit luminal myofibroblast activation.

Our findings may have significant clinical implications. The emergence of luminal myofibroblasts is a major complication in vasculitis that can result in luminal narrowing and arterial occlusion (Orenstein et al, 2012). A majority of KD patients with giant coronary aneurysms develop arterial stenosis (Tsuda et al, 2005), creating a significant risk for myocardial infarction (Friedman et al, 2016) and treatments that prevent adverse vascular remodelling in such high-risk patients are required. Our finding that mTOR signalling is essential for luminal myofibroblast formation suggests this pathway is an ideal therapeutic target to reduce arterial stenosis. Moreover, our findings that mTOR is activated in the luminal myofibroblasts of patients with KD, GCA and TAK, imply that mTOR inhibition may be more widely effective in systemic vasculitis.

Targeting mTOR is a feasible strategy given the availability of safe and effective inhibitors, which are now widely used in drug-eluting stents to prevent re-stenosis (Liu and Sabatini, 2020; Rodriguez-Arias et al, 2020). We found that the pharmacological inhibition of mTOR (with rapamycin) can prevent SMC migration and arterial stenosis in the CAWS model of KD. Our current study also indicates that the mode of action for rapamycin is (at least in part) through inhibiting SMC activation. Hence, we suggest that mTOR is an intrinsic, essential and druggable pathway which is activated in the luminal myofibroblasts that drive adverse blood vessel remodelling in vasculitis patients. We believe these findings provide a compelling rationale for

clinical trials of mTOR inhibitors as a novel therapeutic strategy in systemic vasculitis.

## Methods

### Reagents and tools table

| Reagent/resource | Reference or source | Identifier or catalogue number |
|---|---|---|
| **Experimental models** | | |
| Col1a2^CreERT2 mice | Zheng et al, 2002 | NA |
| Wt.1^CreERT2 mice | Zhou et al, 2008 | NA |
| R26^eYFP | Srinivas et al, 2001 | NA |
| VeCAD^CreERT2 | Monvoisin et al, 2006 | NA |
| Myh11^CreERT2 | Wirth et al, 2008 | JAX strain #019079 |
| Raptor^flox mice | Bentzinger et al, 2008 | NA |
| **Antibodies** | | |
| GFP | Abcam | ab290 |
| Anti-mouse CD31 | BD | Clone MEC13.3 |
| Anti-human α-SMA-AF488 | Invitrogen | Clone 1A4 |
| Anti-mouse CD146 | BD | Clone ME9F1 |
| Anti-mouse PDGFRα | BD | Clone APA5 |
| pS6 | Cell Signaling Technology | Clone D68F8 |
| Ki67 | Invitrogen | Clone SolA15 |
| Anti-human CD146 | OriGene | Clone UMAB155 |
| Anti-mouse CD45.2 | Invitrogen | Clone 104 |
| Anti-mouse podoplanin | Invitrogen | Clone 8.1.1 |
| Donkey-anti-rabbit AF594 | Abcam | ab150064 |
| Donkey-anti-rat AF647 | Abcam | ab150155 |
| **Oligonucleotides and other sequence-based reagents** | | |
| RT-PCR primers | This study | Methods |
| **Chemicals, enzymes and other reagents** | | |
| Tamoxifen | Sigma | T5648 |
| Rapamycin | MedChemExpress | HY-10219 |
| SuperScript III Reverse Transcriptase | Invitrogen | 18080093 |
| Fast SybrGreen Master Mix | Invitrogen | 4385617 |
| Paraformaldehyde | Sigma | 158127 |
| Sucrose | Sigma | S0389 |
| O.C.T. | Tissue-Tek | 4583 |
| TrueBlack Lipofuscin Autofluorescence Quencher | Biotium | 23007 |

| Reagent/resource | Reference or source | Identifier or catalogue number |
|---|---|---|
| Protein Block | Dako | X090930-2 |
| **Software** | | |
| Graphpad prism | https://www.graphpad.com | |
| ImageJ | https://imagej.nih.gov/ij/index.html | |

## Mice, drug administration and the CAWS model of Kawasaki disease

C57BL/6, Col1a2[CreERT2] (Zheng et al, 2002), Wt1[CreERT2] (Zhou et al, 2008), R26[eYFP] (Srinivas et al, 2001), VeCAD[CreERT2] (Monvoisin et al, 2006), Myh11[CreERT2] (Wirth et al, 2008) and Raptor[flox] mice

(Bentzinger et al, 2008) were bred at the Walter and Eliza Hall Institute of Medical Research (WEHI, Australia) under specific pathogen-free conditions. To induce the nuclear translocation of CreERT2, experimental adult mice (including naive controls) were administered 4 mg tamoxifen (dissolved in peanut oil) by oral gavage daily, for 4 consecutive days. The *Candida albicans* water-soluble (CAWS) complex was prepared as previously described (Nagi-Miura et al, 2006; Tada et al, 2008; Stock et al, 2016). To induce a Kawasaki-like disease, adult mice were injected intraperitoneally with either 4 mg of CAWS once or with 3 mg CAWS on 2 consecutive days. For rapamycin treatment, mice were administered rapamycin (8 mg/kg; diluted in 5% Tween80/5% PEG400/H2O; MedChemExpress) by intraperitoneal (IP) injections three times per week (Monday, Wednesday, Friday). All procedures were performed at WEHI and approved by the WEHI Animal Ethics Committee.

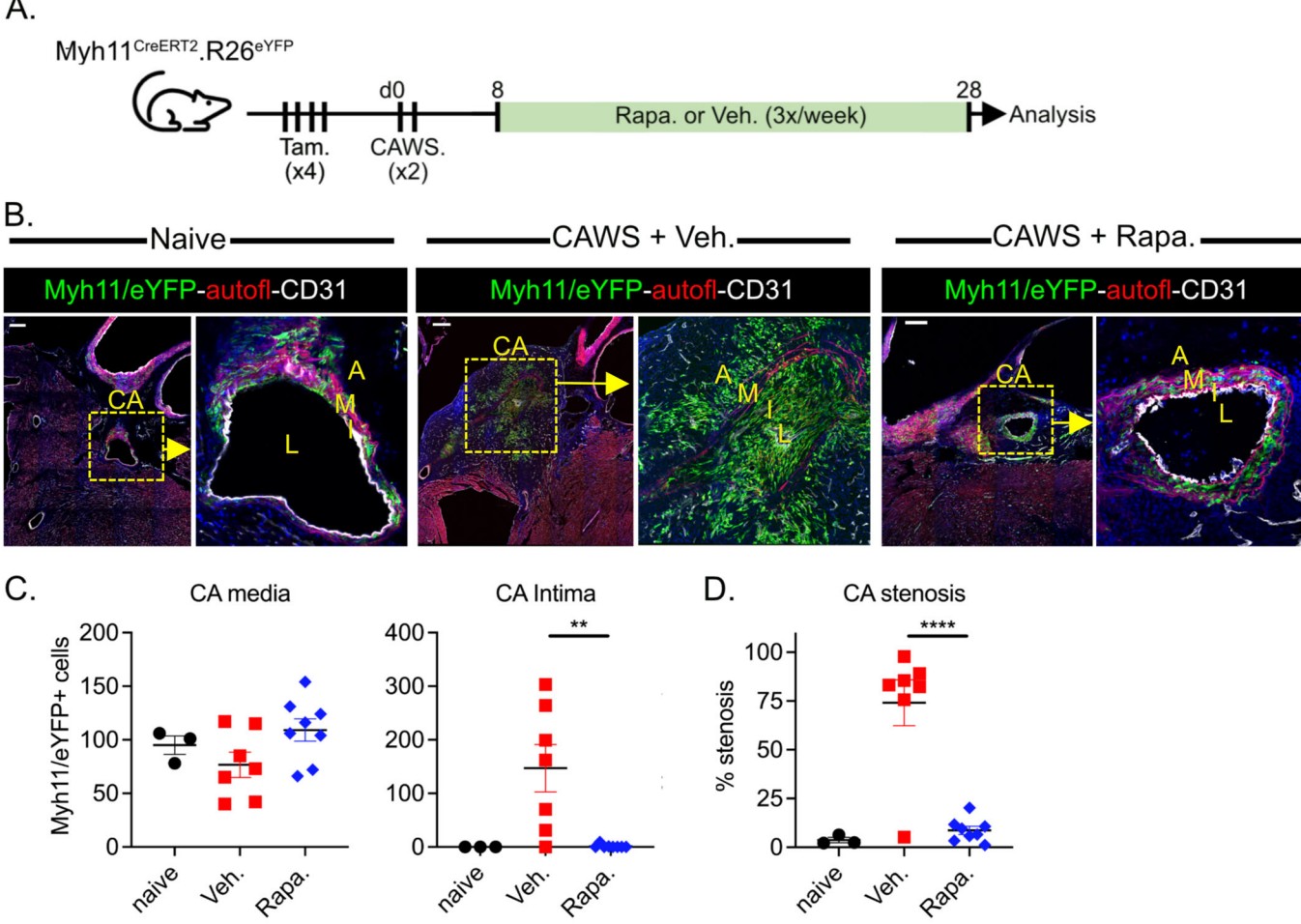

**Figure 7. Pharmacological mTOR inhibition prevents luminal myofibroblast formation and coronary artery stenosis during CAWS-induced vasculitis.**

(A) Experimental schema. (B) Cardiac sections from naive or CAWS-injected Myh11[CreERT2].R26[eYFP] mice treated with Rapamycin of Vehicle control. Sections were stained for GFP to identify Myh11 + /eYFP+ cells (green), CD31 to label endothelial cells (white) and autofluorescence used to identify elastin fibres of the media (red). Graphs show the number of Myh11 + /eYFP+ cells within the coronary artery media or intima (C) and CA stenosis (D). Points depict individual mice (with mean ± SEM) pooled from two independent experiments. The coronary artery (CA), adventitia (A), media (M), intima (I) and lumen (L) are annotated and scale bars are 100 μm. **$P < 0.01$; ****$P < 0.0001$ with two-tailed Student's *t* tests. Exact *P* values (to 4 decimal points) for (C) 0.0036 (**), (D) < 0.0001 (****). Source data are available online for this figure.

## Histology and immunofluorescent analysis

For imaging of mouse tissue, hearts were perfused with PBS and then fixed in 2% paraformaldehyde (4 h on ice), dehydrated in 30% sucrose (12–18 h at 4 °C) and embedded in OCT (Tissue-Tek). Hearts were sectioned (10 µM) in the coronal plane and analysed by histopathology or immunofluorescence microscopy. For histology, sections were stained with H&E or Sirius Red. For immunostaining, cardiac sections were hydrated in PBS, permeabilized with 0.1% Triton-X and non-specific staining blocked with serum (Jackson Immunoresearch), 1% BSA and Protein block (Dako). Sections were stained with primary antibodies against GFP (ab290; Abcam), CD31 (MEC13.3; BD), α-SMA (1A4; Invitrogen), CD146 (ME9F1; BD), PDGFRα (APA5; BD), phosphorylated-S6Ser240/244 Ribosomal Protein S6 (pS6; clone D68F8; Cell Signaling Technology) and Ki67 (SolA15; Invitrogen) before detection (where necessary) with fluorochrome-conjugated secondary antibodies (Invitrogen or Abcam). Slides were counter-stained with DAPI, imaged on a Zeiss LSM-880 Confocal Microscope and analysed with ImageJ software. For cell quantification, eYFP+ cells were counted within a 500 µm$^2$ area of adventitia (for adventitial cells) or by counting the total number of eYFP+ cells within the media and intima of a single coronary artery (for medial and intimal cells) using the cell counter tool in ImageJ. To quantify stenosis, the intimal area (defined by the IEL border) and luminal area (based upon the CD31+ endothelial layer) were measured, and stenosis was defined by the formula % stenosis = 100 – ([luminal area/intimal area] × 100).

For human studies, sections from fatal KD and TAK cases were obtained from the Victorian Institute of Forensic Medicine (VIFM), GCA biopsies were obtained from the Royal Melbourne Hospital and normal cardiac tissue was obtained from the Australian Donation and Transplantation Biobank (ADTB). Paraffin sections (7 µM) were dewaxed and subjected to citrate antigen retrieval. Sections were permeabilized with 0.1% Triton-X, blocked with serum/BSA, True-Black Lipofuscin Autofluorescence Quencher (Biotium) and Protein block (Dako) before staining with anti-pS6 (pS6; clone D68F8; Cell Signaling Technology), CD146 (UMAB155; OriGene) and α-SMA-AF488 (1A4; Invitrogen) primary antibodies for detection (where necessary) with fluorochrome-conjugated secondary antibodies. All procedures were approved by the VIFM, RMH, ADTB and WEHI Human Research Ethics Committee.

## Flow cytometry of cardiac tissue

For flow cytometric analysis, murine hearts were digested in type I collagenase (1 mg/ml; Worthington) with DNase I (10 µg/ml; Sigma). Single-cell suspensions were then stained with directly conjugated mAbs against CD45.2 (104), CD31 (390), podoplanin (8.1.1), PDGFRα/CD140A (APA5) and CD146 (ME9F1) (BD Bioscience, eBioscience or Biolegend). Propidium iodide (100 ng/ml) was added immediately prior to data acquisition on a Fortessa (BD) FACS machine and analysed using Flowjo software.

## Real-time quantitative PCR

For gene expression analysis of purified populations, cells were sorted on FACSAria II (BD), and RNA was extracted using the RNeasy Micro Kit (Qiagen) and cDNA synthesised with Super-Script III Reverse Transcriptase (Invitrogen) using oligo-dT primers (Promega). Quantitative real-time PCR was performed with Fast Sybergren Master mix (Invitrogen) with customised primers for *Hrpt* (forward—CCCTCTGGTAGATTGTCGCTTA; reverse—AGATGCTGTTACTGATAGGAAATTGA), *Col1a1* (forward—GAGCGGAGAGTACTG GATCG; reverse—GCTTCTTT TCCTTGGGGTTC), *Col1a2* (forward—CAGCGAAGAACT CAT-ACAGC; reverse—GACACCCCTTCTACGTTGT), *Col3a1* (forward—ACGTAAGCACTGGTGGACAG; reverse—GGAGGGCCA TAGCTGAACTG), *P4Ha3* forward—ATCCCATGGCTGGAG-GAAGC; reverse—AACTCTCGAGAGAGGCTGAGG), *SAA3* (forward—TCTTGATCCTGGGAGTTGACAGC; reverse—TAGGCTC GCCACATGTCTCT), *SPP1* (forward—ATGAGGCTGCAGTTC TCCTGG; reverse—TATAGGATCTGGGTGCAGGCT), *Ccl2* (forward—TCTCTTCCTCCACCACCATGC; reverse—AGCTTCTT TGGGACACCTGCTG), *Ccl7* (forward—TCTGCCACGCTTCTGT GCC; reverse—AACAGCTTCCCAGGGACACCG). Target gene expression was normalised to *Hrpt* (ΔCT), and relative expression converted by the 2$^{-\Delta CT}$ method.

## In vitro SMC assays

To generate primary murine SMCs, the descending aorta was processed by mechanically removing the adipose and adventitial layers before digestion in type I collagenase/DNase I. SMCs were then cultured in DMEM containing 10% foetal bovine serum and antibiotics to 70% confluence before experimental assays. For pS6 assays, cells were cultured with one of IL-1β, IL-17, Tnf or GM-CSF (20 ng/ml; Peprotech) with or without rapamycin (0.1 µM; MedChemExpress). For SAA3 assays, cells were cultured as above, with the addition of BFA (5 µM; Sigma). After 2 h cytokine stimulation, cells were harvested with trypsin, stained for surface CD146, fixed and permeabilised (BD) and stained for intracellular pS6 (D57.2.2e; CST) and SAA3 (JOR110A; BD) for analysis by flow cytometry.

## Statistical analysis

Statistical analysis was performed with Prism 6.0 (GraphPad Software) using unpaired, two-tailed Student *t* tests. Statistical significance levels are expressed as *$P < 0.05$; **$P < 0.01$; ***$P < 0.001$; ****$P < 0.0001$.

# Data availability

This study includes no data deposited in external repositories.

The source data of this paper are collected in the following database record: biostudies:S-SCDT-10_1038-S44319-024-00251-1.

# Peer review information

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

## Acknowledgements

The authors thank Edan Azzopardi, Tom Kitson and Lauren Wilkins (WEHI, Melbourne, Australia) for outstanding technical assistance and Drs Christine Biben, Felicity Jackling, Jane Visvader (WEHI, Melbourne, Australia) and Axel Kallies (The Peter Doherty Institute, Melbourne, Australia) for the provision of mice. Biospecimens used in this project were provided by the Australian Donation and Transplantation Biobank (ADTB). We acknowledge the contribution of the Victorian Liver Transplant Unit, DonateLife Victoria and the ADTB to this research project. We gratefully acknowledge the generosity of the deceased organ donors and their families in providing invaluable tissue samples. This work was supported by an Arthritis Rheumatology Australia Project Grant, the Reid Charitable Trusts and the Australian National Health and Medical Research Council (NHMRC) Program Grant 1113577 and Clinical Practitioner Fellowship 1154235 (to IPW). This study was made possible through the Victorian State Government Operational Infrastructure Support and the Australian Government National Health and Medical Research Council Independent Research Institute Infrastructure Support scheme. The ADTB was funded by the Australian Centre for Transplant Excellence and Research.

## Author contributions

**Angus T Stock**: Conceptualisation; Data curation; Formal analysis; Funding acquisition; Investigation; Methodology; Writing—original draft; Project administration; Writing—review and editing. **Sarah Parsons**: Resources; Formal analysis; Writing—original draft. **Jacinta A Hansen**: Methodology. **Damian B D'Silva**: Methodology. **Graham Starkey**: Resources. **Aly Fayed**: Resources. **Xin Yi Lim**: Resources. **Rohit D'Costa**: Resources. **Claire L Gordon**: Resources. **Ian P Wicks**: Supervision; Funding acquisition; Writing—original draft; Writing—review and editing.

Source data underlying figure panels in this paper may have individual authorship assigned. Where available, figure panel/source data authorship is listed in the following database record: biostudies:S-SCDT-10_1038-S44319-024-00251-1.

## Disclosure and competing interests statement

The authors declare no competing interests.

# Expanded View Figures

**Figure EV1.   CAWS-induced vasculitis triggers the formation of collagen-expressing luminal myofibroblasts.**

(A) Cardiac sections from naive or CAWS-injected mice (4–5 weeks post injection) analysed by histology or immunofluorescent microscopy. For the latter, sections were stained for CD31 to label endothelial cells and autofluorescence was used to identify elastin fibres of the media. (B) Graphs show the maximum width of the coronary artery intima for individual mice (with mean ± SEM) pooled from 3 independent experiments. (C) Genetic schema for Col1a2$^{CreERT2}$.R26$^{eYFP}$ system. (D) Flow cytometric analysis of cardiac cells from Col1a2$^{CreERT2}$.R26$^{eYFP}$ (and control) mice showing endogenous eYFP+ expression and PDGFRα staining. Inset value is the mean of 4–6 mice from 3 independent experiments. (E) Experimental schema. (F) Cardiac sections from naive or CAWS-injected Col1a2$^{CreERT2}$.R26$^{eYFP}$ mice stained for GFP to identify Col1a2 + /eYFP+ cells (green), CD31 to label endothelial cells (white) and autofluorescence to identify elastin fibres of the media (red). (G) Graphs show the number of Col1a2 + /eYFP+ cells within each vessel layer for individual mice (with mean ± SEM) pooled from 3 independent experiments. The coronary artery (CA), adventitia (A), media (M) and intima (I) are annotated and scale bars are 1000 μm. *** $P < 0.001$ with two-tailed Student's *t* tests. Exact *P* values (to 4 decimal points) for: (B) 0.0004 (***). Source data are available online for this figure.

◀

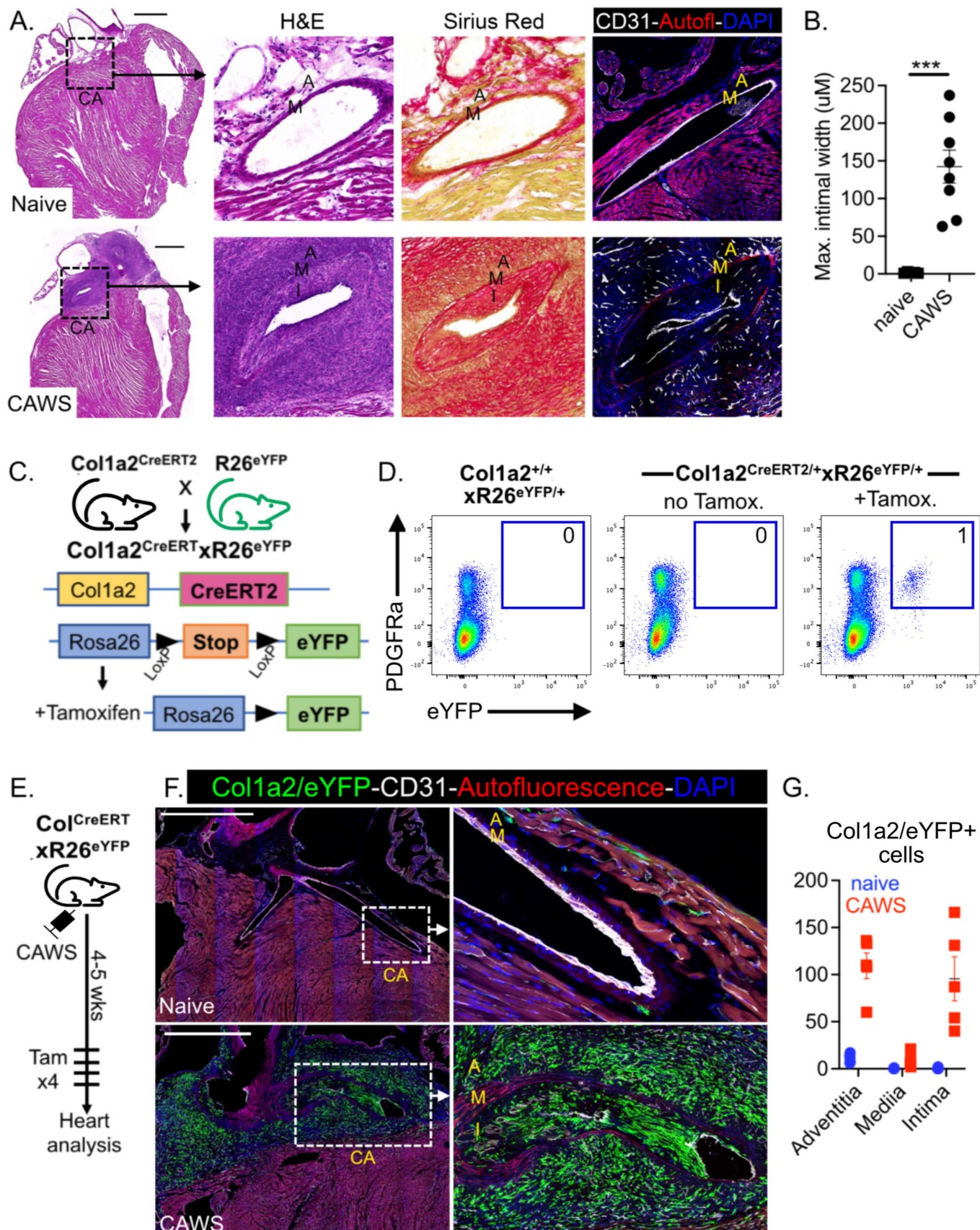

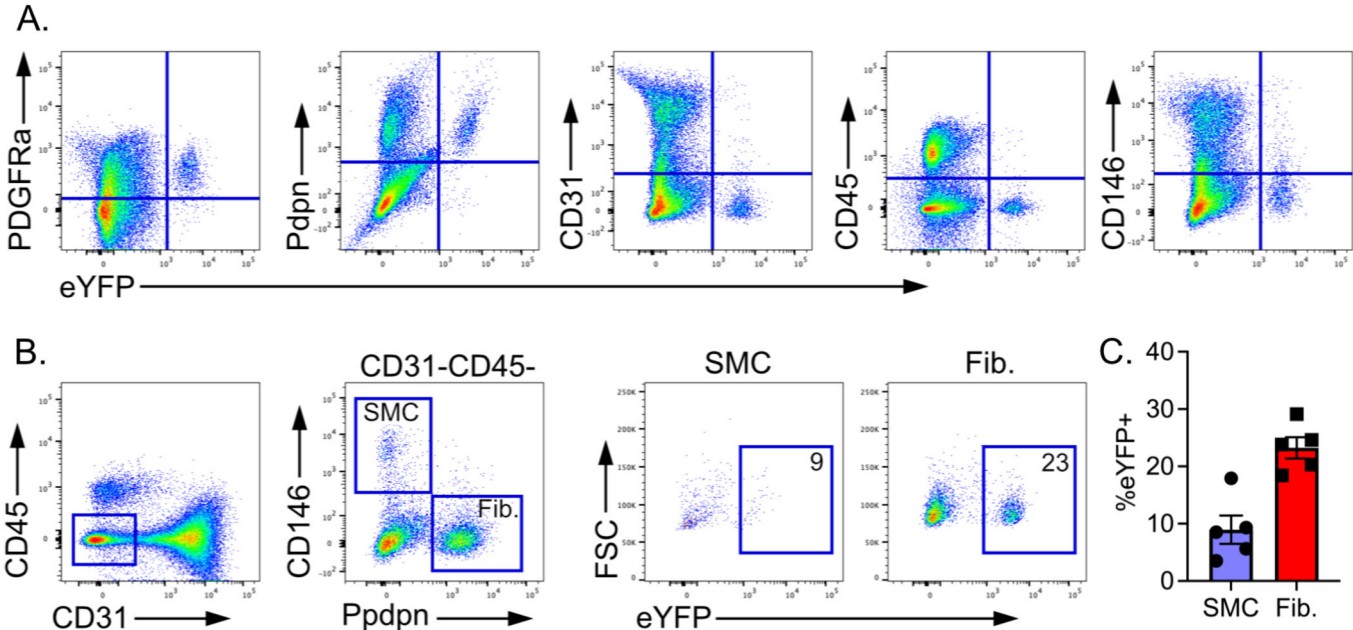

**Figure EV2. E10.5 Wt1 recombination predominantly labels resident cardiac fibroblasts.**

(**A**) Flow cytometric analysis of cardiac cells from Wt1<sup>CreERT2</sup>.R26<sup>eYFP</sup> mice (administered tamoxifen at E10.5) showing endogenous eYFP+ expression versus lineage-specific markers. (**B**) Gating strategy and eYFP labelling of SMC (CD45-CD31-CD146 + ) and fibroblasts (CD45-CD31-Pdpn + ). (**C**) Graphs show the proportion of eYFP labelling in SMCs and fibroblasts for individual mice (with mean ± SEM) pooled from 2 independent experiments. Source data are available online for this figure.

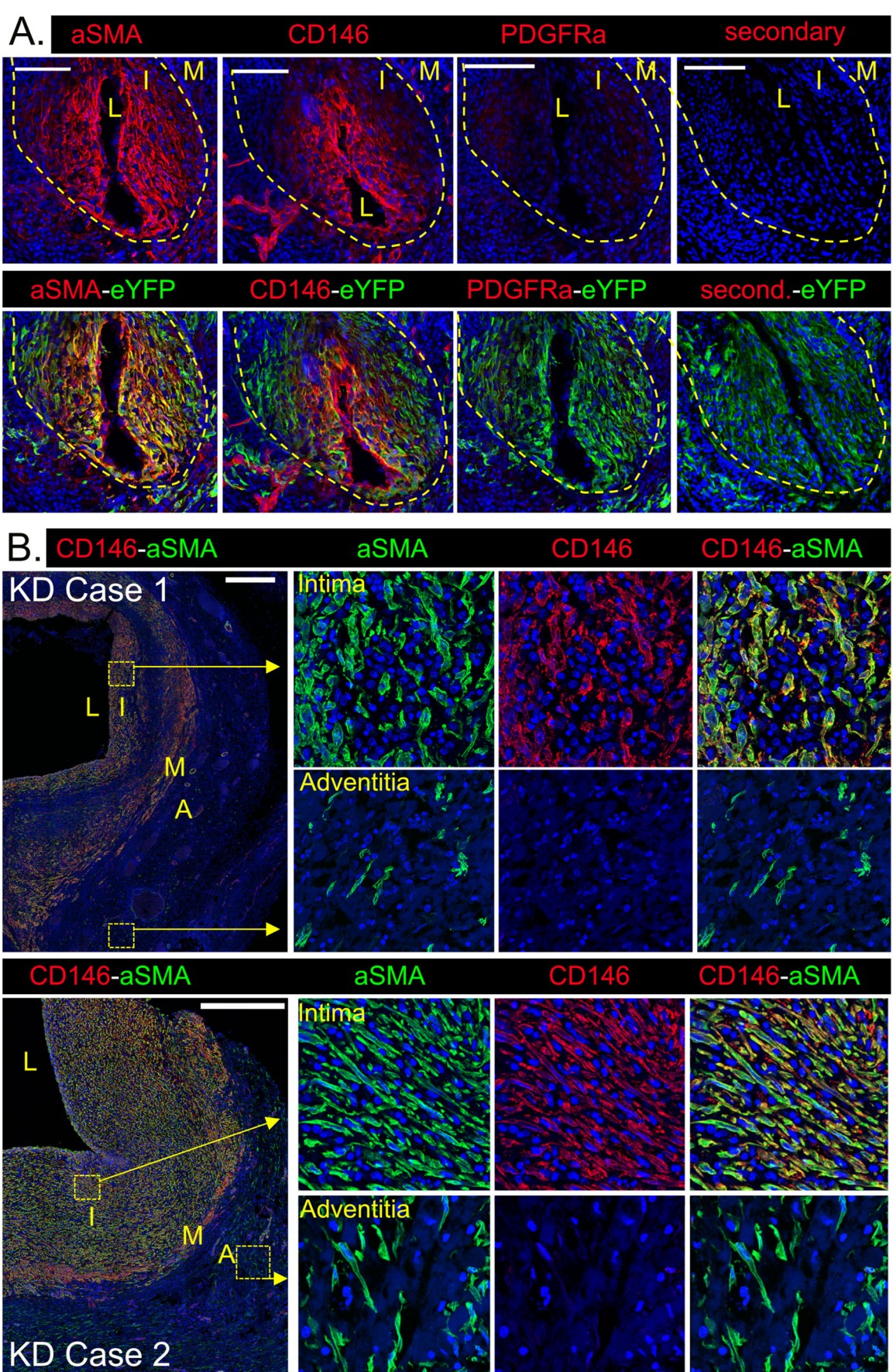

◀ **Figure EV3.  Luminal myofibroblasts express mural cell markers.**

(A) Cardiac sections from CAWS-injected Myh11[CreERT2].R26[eYFP] mice (4–5 weeks post CAWS) analysed by immunofluorescent microscopy. Sections were stained for GFP to identify Myh11 + /eYFP+ cells (green) and either α-SMA, CD146 or PDGFRα (red). Scale bars are 100 µm. (B) Coronary artery sections from two acute KD fatalities stained for α-SMA (green) and CD146 (red) and analysed by confocal microscopy. Scale bars are 1000 µm. The adventitia (A), media (M), intima (I) and lumen (L) are annotated throughout. Source data are available online for this figure.

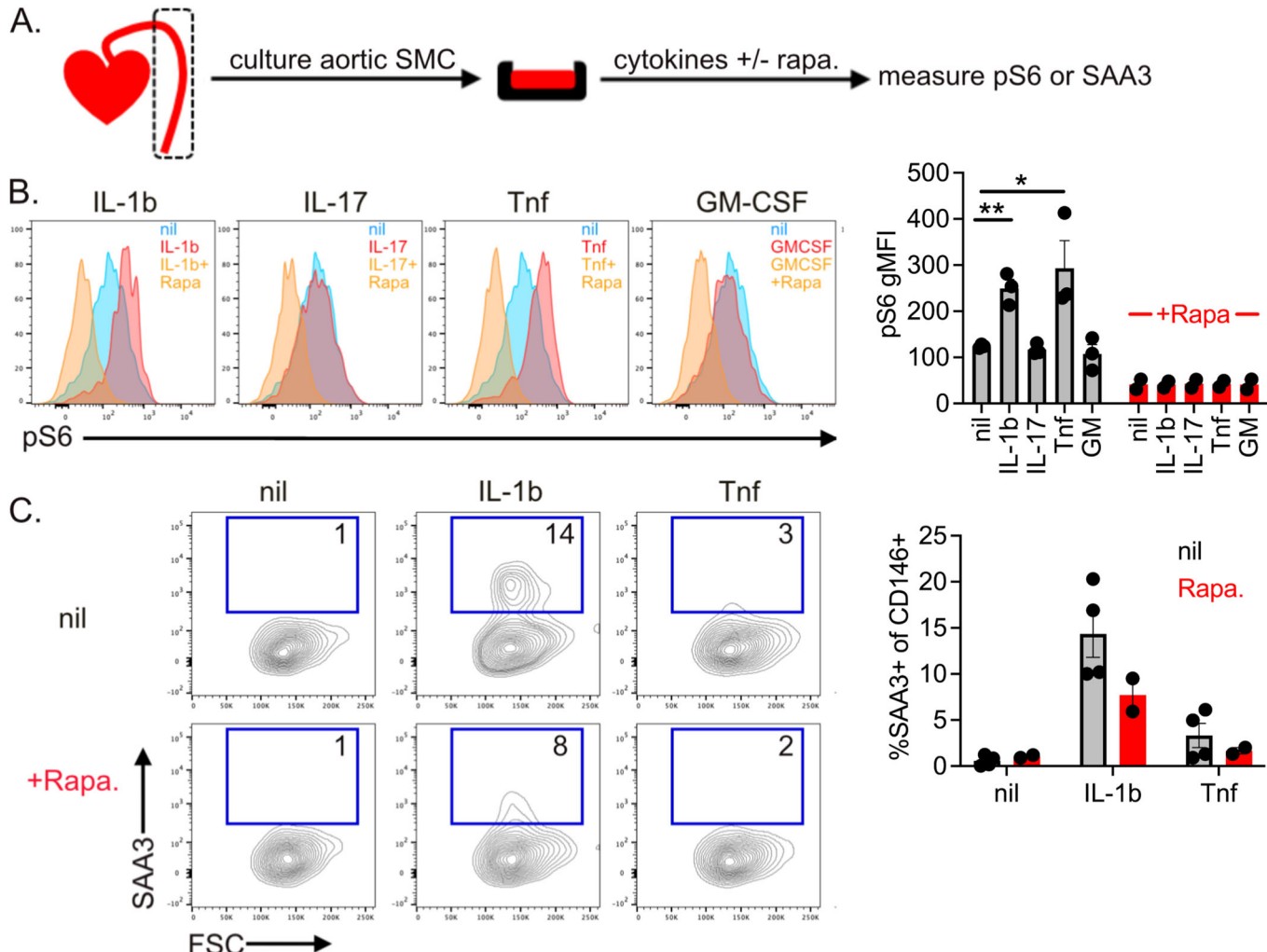

**Figure EV4.   IL-1β activates mTOR in SMCs in vitro.**

(A) Experimental schematic. (B, C) FACS plots are gated on CD146+ cells and show intracellular staining for pS6 (B) or SAA3 (C) in cultured SMCs stimulated with cytokines $+/-$ rapamycin. Graphed points depict individual lines (with mean ± SEM) pooled from 2 independent experiments. *$P < 0.05$; **$P < 0.01$ with two-tailed Student's $t$ tests. Exact $P$ values (to 4 decimal points) for (B) IL-1β 0.0032 (**), Tnf 0.0482 (*). Source data are available online for this figure.

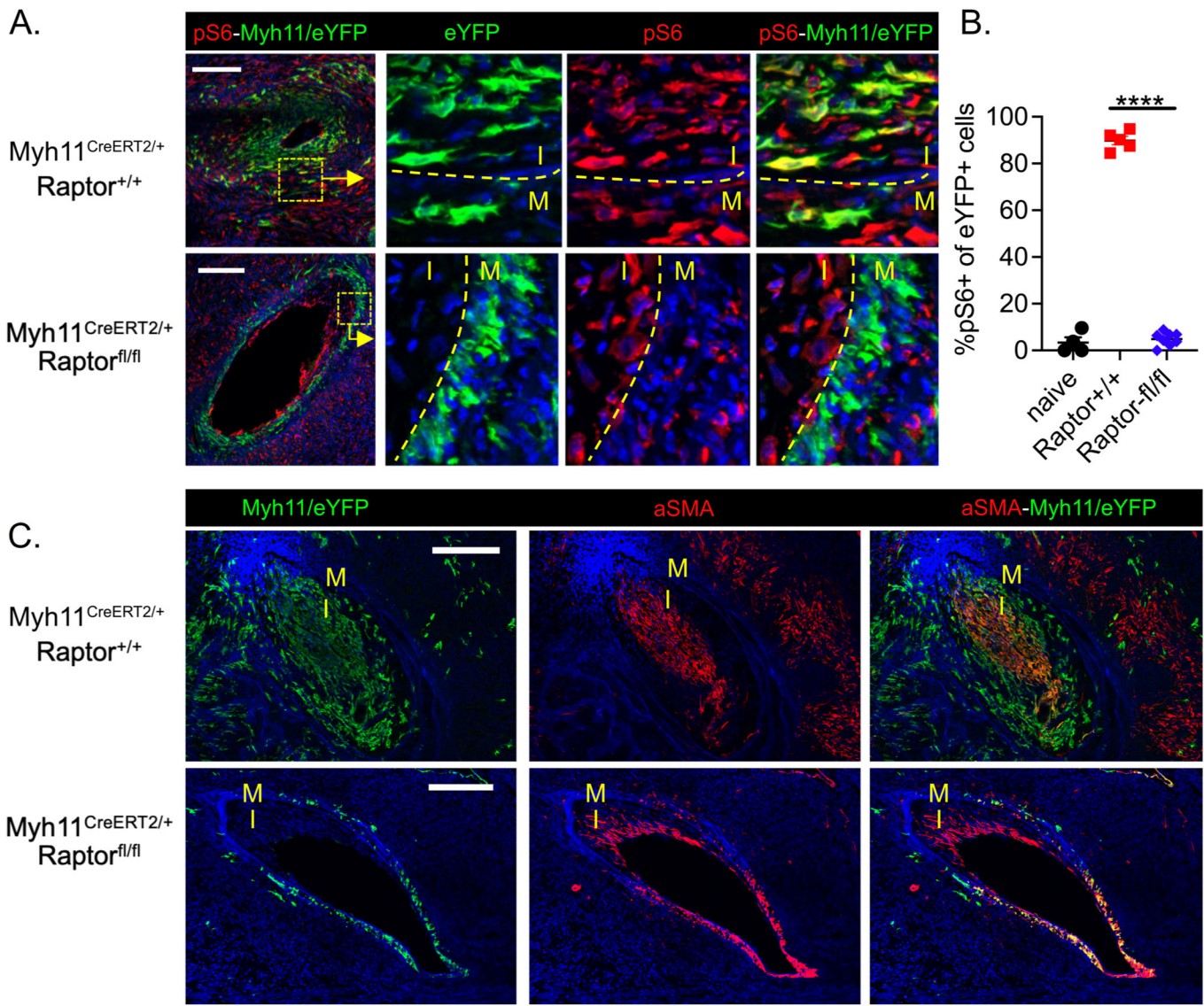

**Figure EV5. Analysis of Raptor-deficient SMCs.**

(A–C) Representative cardiac sections from CAWS-injected Myh11$^{CreERT2}$.Raptor$^{+/+}$.R26$^{eYFP}$ or Myh11$^{CreERT2}$.Raptor$^{fl/fl}$.R26$^{eYFP}$ mice (4–5 weeks post injection). (A, B) Sections are stained for GFP to identify Myh11 + /eYFP+ cells (green) and pS6 (red) to assess mTOR signalling. Graphs depict the % eYFP+ cells that are pS6+ for individual mice (with mean ± SEM) pooled from 3 independent experiments. (C) Sections show eYFP+ cells and α-SMA expression. The IEL is shown as a dashed line in (A) and the adventitia (A), media (M), intima (I) and lumen (L) are annotated throughout. Scale bars are 100 μm. ****$P < 0.0001$ with two-tailed Student's $t$ tests. Exact $P$ values (to 4 decimal points) for (B) < 0.0001 (****). Source data are available online for this figure.

