## [Peer Review File · EMBO Reports]

mTOR signalling controls the formation of smooth muscle cell-derived luminal myofibroblasts during vasculitis

Angus Stock, Sarah Parsons, Jacinta Hansen, Damian D'Silva, Graham Starkey, Aly Fayed, Xin Yi Lim, Rohit D'Costa, Claire Gordon, and Ian Wicks

Corresponding author(s): Angus Stock (stock.a@wehi.edu.au) , Ian Wicks (wicks@wehi.edu.au)

Review Timeline:

Submission Date:	19th Dec 23
Editorial Decision:	15th Feb 24
Revision Received:	2nd Jul 24
Editorial Decision:	31st Jul 24
Revision Received:	2nd Aug 24
Accepted:	20th Aug 24

Editor: Deniz Senyilmaz Tiebe

Transaction Report:

Dear Dr. Stock,

Thank you for the submission of your research manuscript to our journal, which was now seen by three referees, whose reports are copied below.

My apologies for this unusual delay in getting back to you. It took longer than anticipated to receive the full set of referee reports.

Referees express interest in the proposed role of mTOR signaling in formation of smooth muscle cell-derived intimal fibroblasts in the context of vasculitis. However, they also raise significant concerns that need to be addressed to consider publication here.

Given these positive recommendations, we would like to invite you to submit a revised manuscript. Please revise your manuscript with the understanding that the referee concerns (as in their reports) must be fully addressed and their suggestions taken on board. Please address all referee concerns in a complete point-by-point response. Acceptance of the manuscript will depend on a positive outcome of a second round of review. It is EMBO reports policy to allow a single round of major experimental revision only and acceptance or rejection of the manuscript will therefore depend on the completeness of your responses included in the next, final version of the manuscript.

We realize that it is difficult to revise to a specific deadline. In the interest of protecting the conceptual advance provided by the work, we recommend a revision within 3 months. Please discuss the revision progress ahead of this time with me if you require more time to complete the revisions, or if you have questions or comments regarding the revision (also by video chat).

1. A data availability section providing access to data deposited in public databases is missing (where applicable).
2. Your manuscript contains statistics and error bars based on $n=2$. Please use scatter plots in these cases.

You can submit the revision either as a Scientific Report or as a Research Article. For Scientific Reports, the revised manuscript can contain up to 5 main figures and 5 Expanded View figures, and it should not exceed 27000 characters. If the revision leads to a manuscript with more than 5 main figures it will be published as a Research Article. In this case the Results and Discussion section should be separate. If a Scientific Report is submitted, these sections have to be combined. This will help to shorten the manuscript text by eliminating some redundancy that is inevitable when discussing the same experiments twice. In either case, all materials and methods should be included in the main manuscript file.

4) a .docx formatted letter INCLUDING the reviewers' reports and your detailed point-by-point responses to their comments. As part of the EMBO publication's Transparent Editorial Process, EMBO reports publishes online a Review Process File (RPF) to accompany accepted manuscripts. This File will be published in conjunction with your paper and will include the referee reports, your point-by-point response and all pertinent correspondence relating to the manuscript. <https://www.embopress.org/page/journal/14693178/authorguide#transparentprocess>

5) a complete author checklist, which you can download from our author guidelines <https://www.embopress.org/page/journal/14693178/authorguide>. Please insert information in the checklist that is also reflected in the manuscript. The completed author checklist will also be part of the RPF.

6) Please note that all corresponding authors are required to supply an ORCID ID for their name upon submission of a revised manuscript (<<https://orcid.org/>>). Please find instructions on how to link your ORCID ID to your account in our manuscript tracking system in our Author guidelines <<https://www.embopress.org/page/journal/14693178/authorguide#authorshipguidelines>>

7) Before submitting your revision, primary datasets produced in this study need to be deposited in an appropriate public database (see <https://www.embopress.org/page/journal/14693178/authorguide#datadeposition>). Please remember to provide a reviewer password if the datasets are not yet public. The accession numbers and database should be listed in a formal "Data Availability" section placed after Materials & Method (see also <https://www.embopress.org/page/journal/14693178/authorguide#datadeposition>). Please note that the Data Availability Section is restricted to new primary data that are part of this study. * Note - All links should resolve to a page where the data can be accessed. *
If your study has not produced novel datasets, please mention this fact in the Data Availability Section.

Additional information on source data and instruction on how to label the files are available:
<https://www.embopress.org/page/journal/14693178/authorguide#sourcedata>

9) Our journal encourages inclusion of *data citations in the reference list* to directly cite datasets that were re-used and obtained from public databases. Data citations in the article text are distinct from normal bibliographical citations and should directly link to the database records from which the data can be accessed. In the main text, data citations are formatted as follows: "Data ref: Smith et al, 2001" or "Data ref: NCBI Sequence Read Archive PRJNA342805, 2017". In the Reference list, data citations must be labeled with "[DATASET]". A data reference must provide the database name, accession number/identifiers and a resolvable link to the landing page from which the data can be accessed at the end of the reference. Further instructions are available at <http://www.embopress.org/page/journal/14693178/authorguide#referencesformat>

- the name of the statistical test used to generate error bars and P values,
- the number (n) of independent experiments (please specify technical or biological replicates) underlying each data point,
- the nature of the bars and error bars (s.d., s.e.m.),
- If the data are obtained from n Program fragment delivered error `Can't locate object method "less" via package "than" (perhaps you forgot to load "than"?) at //ejpvfs23/sites23b/embor_www/letters/embor_decision_revise_and_review.txt line 56.' 2, use scatter blots showing the individual data points.

12) Please also note our reference format:

I look forward to seeing a revised version of your manuscript when it is ready. Please let me know if you have questions or comments regarding the revision.

Kind regards,

Deniz Senyilmaz Tiebe

Deniz Senyilmaz Tiebe, PhD
Scientific Editor
EMBO Reports

Referee #1:

In this study, Stock et al. perform lineage cellular tracing in a mouse model of Kawasaki disease (KD) vasculitis to investigate the origin of "intimal fibroblasts" which proliferate in the lumen of inflamed coronary arteries of mice injected with candida albicans water-soluble extract (CAWS). They report that Col1a2 expressing cells are detected in inflamed coronaries of mice that received CAWS, indicating a possible fibroblast origin. They ruled out an endothelial and adventitial origin and identified a media to intimal cellular migration of smooth muscle cells (Myh11) associated with proliferation, a pro-fibrotic cellular program, and higher expression of pS6, as a marker of mTOR activity. They observed a similar increase in pS6 in α SMA+ cells on tissue sections collected from patients with KD, Takayasu, and giant cells arteritis. Furthermore, they show that in mice, blocking the mTOR pathway (RAPTOR^{fl/fl}) in Myh11+ cells did not impact the overall severity of CAWS cardiac inflammation but reduced intimal myofibroblast proliferation in the coronary arteries. Overall, this is a well-written and conducted study involving mouse models and human tissues on an interesting topic. However, while the adventitial/medial SMCs origin of luminal myofibroblast is not novel per se, the reported higher expression of pS6 in human samples and mice is novel, as well as blocking RAPTOR in these cells decreases intimal myofibroblast proliferation. I recommend this study for publication with some modifications.

Major comments:

1) The authors call the cells migrating to the lumen of the coronary artery "intimal fibroblasts," "fibroblasts," and in some rare instances: "myofibroblasts." This leads to some confusion. It is an accepted consensus that in the context of KD vasculitis, "luminal myofibroblastic proliferation," or LMP, is a pathological KD stage and that proliferating "myofibroblasts" arise from the adventitial/media smooth muscle cells (SMCs) (Orenstein et al. Plos One 2012 and McCrindle Circulation 2017). Is there a particular reason the authors call these cells "intimal fibroblasts" instead of "myofibroblasts"? They should be consistent throughout the manuscript.

2) In the context of vascular inflammation, it is known that vascular smooth muscle cells (SMCs) are highly phenotypic. At some point, vascular SMCs may become "synthetic," characterized by less contractility, increased production of pro-inflammatory mediators, higher proliferation and migration, and acquisition of a fibroblast-like phenotype. At lines 212 to 214, the authors state:

"These findings illustrate that vasculitis induces SMCs to undergo media-to-intimal migration and acquire a pro-fibrotic transcriptional program, resulting in the formation of intimal fibroblasts."

This observation has recently been published with another model; the Lactobacillus Casei cell wall extract (LCWE) model of KD vasculitis (Porritt et al. Circulation Research 2021 and Zhang Y. Circulation Research 2020) and has been shown to be dependent on IL-1 β and a "platelet-miR223 axis". The manuscript would benefit from discussing this concept in the context of these earlier studies.

3) It would be very interesting to know what is upstream of mTOR activation in this KD vasculitis model. The authors correctly mention in the discussion that IL-1 β and/or Notch signaling may be drivers of mTOR activation. They state that it is possible that the IL-1 β -mTOR axis triggers the formation of SMC-derived "intimal fibroblasts". Indeed, a study mentioned in the manuscript (ref 42; Porritt et al. Circulation Research 2021) using the LCWE-induced KD murine model showed the role of IL-1 β in VSMC transition into myofibroblast.

Since the novelty here is the mTOR pathway involvement in myofibroblasts, it would be interesting to investigate if in vitro the stimulation of primary vascular SMCs cultures or isolated vascular SMCs from mice with IL-1 β triggers the mTOR pathway and their transition to synthetic vascular SMCs. Would this be blocked in vascular SMCs isolated from Myh11Cre Raptor^{fl/fl} mice or vascular SMCs treated with an mTOR inhibitor?

5) In the experiments involving Tamoxifen, are the naïve mice injected with Tamoxifen? Does tamoxifen affect the vasculitis in their model?

Minor comments:

5) When the number of cells is calculated with the different strains of mice of use, it seems the scale varies from one graph to another (# of cells /500µm², # cells per 2mm²), and in some graphs, this is not indicated, leading to some confusion and difficulties to compare across figures.

6) line 362, "?" can be deleted.

Referee #2:

Summary

This article addresses the important issue of the origin of intimal fibroblasts, responsible for intimal thickening. Using a previously-established KD model, the authors perform several lineage tracing experiments to find that intimal fibroblasts derive from vascular smooth muscle cells rather than adventitial fibroblasts or endothelial cells. Furthermore, by conditional deletion experiments, they demonstrate that mTOR signaling is essential for this vascular smooth muscle cell-intimal fibroblast transition. There are some limitations with the markers used to identify fibroblasts (e.g. Podn) and the quantification of this labelling (FACS plots showing very few reporter-labelled fibroblasts). Furthermore, I feel that the important issue of neural crest contribution to vascular smooth muscle in the areas of interest in this study should at least be discussed. Also, the identify of intimal fibroblasts should be more carefully investigated, notably in terms of fibroblast lineage markers e.g. show that VSMC-derived fibroblasts express fibroblast lineage markers such as PDGFR α , or are they a subset of VSMCs or intermediate cell-type? More data could be in Supplemental data (e.g. Figs 6&7), and overall figure quality/illustrations should be improved.

The use of multiple Cre lines has enabled the authors to provide convincing evidence that intimal fibroblasts derived from the VSMC lineage, but not fibroblast or endothelial cells. The study provides original insight into the process of intimal thickening, with significant clinical implications, that will be of interest to the cardiovascular field.

Specific comments

1. The study focuses on proximal coronary artery, and there is no discussion on the heterogeneity of vascular smooth muscle cells. Indeed, VSMCs of proximal coronary arteries have been shown to derive from the neural crest, whereas more distal VSCMs are derived from epicardium (Jian et al. Development 2000 <https://doi.org/10.1242/dev.127.8.1607>). Performing neural crest-specific lineage tracing (e.g. Wnt1-Cre) would provide important insight into the lineage and cell-type origin of intimal fibroblasts of the proximal CoA.

2. Figure 1: The data in this figure is not particularly novel or useful, the model has already been described, and the fact that intimal fibroblasts express collagen is not surprising. This could be put into supplemental data. Furthermore, the usefulness of the FACS experiment is questionable considering the very low percentage of fibroblast labelled. Immunofluorescence quantifications are used throughout the manuscript, hence demonstration of co-labeling of fibroblasts in the areas of interest would be more appropriate. Finally, the choice of fibroblast marker is atypical, in particular for a manuscript focused on fibroblasts. The authors should use more robust makers such as PDGFR α , and use these in IF experiments.

3. Figures 2-3: Based on the images/data provided, one wonders whether the col1a2CreERT2 mouse would not be a better line for lineage tracing resident fibroblasts than Wt1CreERT2? Concerning Wt1CreERT2, early induction (E10.5) leads to epicardial labelling, and hence EPCD labelling in adult heart, including vascular smooth muscle cells and pericytes. This must be discussed. The fact that the authors see adventitial fibroblast labelling, but not VSMC labelling in proximal CoA again raises the issue of the origin of the VSMCs, that must at least be discussed in the manuscript.

4. Figure EV1 also is not particularly informative. Authors state: "Moreover, RT-PCR analysis on sorted cardiac populations revealed that eYFP⁺ cardiac cells from Wt.1CreERT2/+xR26eYFP/+ mice express the highest levels of Col1a2 mRNA (Fig EV1), confirming their identity as resident fibroblasts". Does this mean the Podn⁺ cell population, with lower Col1a2 expression, may not be fibroblasts? Co-labeling of Wt1CreERT2-lineage traced cells with fibroblast markers such as PDGFR in sections is sufficient. What is the proportion of intimal fibroblasts labelled?

5. Figure 8. The effect in the conditional knockout is striking. Could the authors check for lack of pS6 positivity and proliferation in VSMCs in the cKOs? What is the "infiltrate"? The authors should perform immunostaining e.g. CD45/PDGFR α ... to clearly establish whether or not these cells are fibroblasts.

Minor comments:

Check all insets e.g. Figure 4 F, the second inset in CAWS is not positioned correctly.

Improve the appearance of the figures e.g. the illustrations representing mice are not very clear/useful.

Referee #3:

Using different recombinases to selectively hit distinct cell types of the coronary vascular wall, Stock and colleagues performed an elegant lineage tracing study assessing the source of intimal fibroblasts during vasculitis. Using VE-Cadherin-CreERT2 authors showed that endoMT is not contributing to intimal fibroblasts during CAWS-induced vasculitis. Using WT1-CreERT2 authors showed that existing adventitial fibroblasts do expand significantly post-injury and participate extensively in the localized tissue remodeling process, without colonizing the space between the media and the endothelial layer. After ruling-out these two major cell types, authors used a Myh11-CreERT2 to show that the majority of intimal fibroblasts post-CAWS were actually derived from vascular smooth muscle cells (vSMCs). Mechanistically, authors showed that mTOR is essential for the vSMC-to-fibroblast switch, as this phenotypic transition was disrupted in conditional KOs of Raptor driven by Myh11-CreERT2. Enhancing the translational relevance of their findings, authors also showed evidence of increased mTOR signaling in intimal fibroblasts of patients suffering from distinct vascular diseases (Kawasaki disease, Takayasu's arteritis and Giant Cell arteritis). The topic is highly relevant and data are of very good quality and support major conclusions, therefore this work deserves to be published upon minor revisions.

Major comments:

1. Authors used WT1-CreERT2 with administration of tamoxifen at an early embryonic timepoint (E10.5) to label the embryonic epicardium and fibroblasts derived from this pool of cardiac progenitors. However, it is well known that, in addition to fibroblasts, epicardial progenitors also give rise to vascular smooth muscle cells in the coronary vasculature, so it is rather intriguing that authors do not see labeling of vSMCs by WT1-CreERT2.

2. Authors should include in their discussion a sentence clearly stating that their evaluation of the contribution of pre-existing adventitial fibroblasts might be an underestimation, as the WT1-CreERT2 allele does not comprehensively label all cardiac fibroblasts. In addition to the epicardium, during cardiogenesis, fibroblasts can also be derived from other sources, such as the endocardium or, less often, neural crest (Moore-Morris, JCI 2014). Therefore, it is conceivable that pre-existing fibroblasts not labeled by WT1-CreERT2 might contribute to intimal fibroblasts.

3. Authors provided compelling genetic evidence that mTOR is important for production of intimal fibroblasts from vSMCs. However, given the availability of mTOR inhibitors and the fact that the same group has reported that rapamycin prevents adverse vascular remodeling in the CAWS model, it would be interesting to assess the fate of Myh11-CreERT2 lineage-traced cells in mice receiving rapamycin treatment. Is this process as robustly blocked as it was in the conditional KO model?

4. The manuscript would benefit from a diagram summarizing the most relevant conclusions of the work in the last figure (somehow similar to a graphical abstract). This should highlight the contribution of vSMCs to intimal fibroblasts, but also the contribution of adventitial fibroblasts to all the reactive tissue that externally surrounds the media.

Minor comments:

5. Line 105: "To report Cre expression, Col1a2CreERT2 mice were crossed to the R26-stop-eYFP mice to create Col1a2CreERT2 x R26eYFP mice". A reference to the manuscript originally describing the R26eYFP mice should be included here.

6. Microscopy images are mostly of very good quality, however, the low magnification stitchings are, sometimes, of lower quality, with a very strong grid pattern.

Dear Deniz,

Thank you for considering our manuscript for publication at *EMBO reports*. Below are the responses (in blue) to the Referee's comments. For clarity, sections added to the revised manuscript have been underlined in the resubmission. We look forward to hearing your thoughts on the revised study.

Kind Regards,
Angus Stock and Ian Wicks.

Referee#1:

Major Comments -

1) The authors call the cells migrating to the lumen of the coronary artery "intimal fibroblasts," "fibroblasts," and in some rare instances: "myofibroblasts." This leads to some confusion. It is an accepted consensus that in the context of KD vasculitis, "luminal myofibroblastic proliferation," or LMP, is a pathological KD stage and that proliferating "myofibroblasts" arise from the adventitial/media smooth muscle cells (SMCs) (Orenstein et al. Plos One 2012 and McCrindle Circulation 2017). Is there a particular reason the authors call these cells "intimal fibroblasts" instead of "myofibroblasts"? They should be consistent throughout the manuscript.

Thank you, to be consistent with the field, we now refer to the fibroblasts that infiltrate the inflamed coronary artery intima during vasculitis as luminal myofibroblasts throughout the revised manuscript (including the title).

2) In the context of vascular inflammation, it is known that vascular smooth muscle cells (SMCs) are highly phenotypic. At some point, vascular SMCs may become "synthetic," characterized by less contractility, increased production of pro-inflammatory mediators, higher proliferation and migration, and acquisition of a fibroblast-like phenotype. At lines 212 to 214, the authors state:

"These findings illustrate that vasculitis induces SMCs to undergo media-to-intimal migration and acquire a pro-fibrotic transcriptional program, resulting in the formation of intimal fibroblasts."

This observation has recently been published with another model; the Lactobacillus Casei cell wall extract (LCWE) model of KD vasculitis (Porritt et al. Circulation Research 2021 and Zhang Y. Circulation Research 2020) and has been shown to be dependent on IL-1 β and a "platelet-miR223 axis". The manuscript would benefit from discussing this concept in the context of these earlier studies.

Thank you for these observations. We have now included a description of these studies in the revised manuscript. Specifically, we have added the following section to the discussion.

Our findings are consistent with earlier studies using the Lactobacillus Casei cell wall extract (LCWE) model of KD showing that SMCs undergo a phenotypic switch during vasculitis associated with the upregulation of fibrotic and proliferative markers and the increased expression of MMPs and inflammatory cytokines/chemokines (Zhang et al., 2020, Porritt et al., 2021). Critically, these earlier studies demonstrate that inhibiting SMC activation (or dedifferentiation) reduced the severity of cardiac vasculitis and medial thickening (Porritt et al., 2021, Zhang et al., 2020). Our findings that inhibiting SMC activation significantly attenuated coronary artery stenosis are consistent with these earlier studies (Zhang et al., 2020, Porritt et al., 2021), which together provide proof-of-principle that that limiting SMC activation can improve clinical outcomes in KD.

3) It would be very interesting to know what is upstream of mTOR activation in this KD vasculitis model. The authors correctly mention in the discussion that IL-1 β and/or Notch signaling may be drivers of mTOR activation. They state that it is possible that the IL-1 β -mTOR axis triggers the formation of SMC-derived "intimal fibroblasts". Indeed, a study mentioned in the manuscript (ref 42; Porritt et al. Circulation Research 2021) using the LCWE-induced KD murine model showed the role of IL-1 β in VSMC transition into myofibroblast.

Since the novelty here is the mTOR pathway involvement in myofibroblasts, it would be interesting to investigate if in vitro the stimulation of primary vascular SMCs cultures or isolated vascular SMCs from mice with IL-1 β triggers the mTOR pathway and their transition to synthetic vascular SMCs. Would this be blocked in vascular SMCs isolated from Myh11Cre Raptorfl/fl mice or vascular SMCs treated with an mTOR inhibitor?

We thank the reviewer for this suggestion and have performed extra experiments and included the data as Fig EV4 in the revised manuscript. As predicted by the reviewer, we found that in vitro, IL-1 β triggered mTOR signalling (as measured by pS6) and effector functions (as measured by SAA3 production) in SMCs, which were reduced by the mTOR inhibitor rapamycin. These results are consistent with IL-1 β being a potential upstream regulator of mTOR during vasculitis and are described in the results section of the revised manuscript as follows:

IL-1 β activates mTOR signalling and SMC dependent effector functions:

We explored what upstream signals may drive mTOR activation in SMCs. We tested IL-1 β , IL-17, Tnf and GM-CSF, which are cytokines known to have critical roles in driving cardiac vasculitis in KD (Lee et al., 2015, Lin et al., 2024, Stock et al., 2016, Stock et al., 2019). For these experiments, we treated murine aortic SMCs with cytokines (with or without the mTOR inhibitor rapamycin) and stained for phospho-S6 and SAA3, a cytokine we found to be highly expressed by activated SMCs during CAWS induced vasculitis (Fig. 2H). We found that both IL-1 β and Tnf (but not IL-17 or GM-CSF) triggered mTOR signalling, as revealed by increased pS6 (Fig. EV4B). Moreover, we found that IL-1 β (but not Tnf) induced SAA3 expression by SMCs, which was reduced by rapamycin (Fig. EV4C). These findings show that IL-1 β can directly activate mTOR signalling in SMCs and drive effector functions (i.e. SAA3 production). Consistent with an earlier study (Porritt et al., 2021), these findings raise the possibility that an IL-1-mTOR axis promotes SMC activation.

5) In the experiments involving Tamoxifen, are the naïve mice injected with Tamoxifen? Does tamoxifen affect the vasculitis in their model?

Yes, naïve mice were administered tamoxifen. We have amended the methods section of the revised manuscript (as follows) to clarify this point.

To induce the nuclear translocation of CreERT2, experimental adult mice (including naïve controls) were administered 4mg tamoxifen (dissolved in peanut oil) by oral gavage daily, for four consecutive days.

From previous experience with this model, we have no reason to suspect that tamoxifen affects vasculitis.

Minor comments:

5) When the number of cells is calculated with the different strains of mice of use, it seems the scale varies from one graph to another (# of cells /500 μ m², # cells per 2mm²), and in some graphs, this is not indicated, leading to some confusion and difficulties to compare across figures.

We agree and have amended the revised manuscript to enumerate cells within a 500 μ m² area (for adventitial cells) throughout. We have added a description to the methods section of the revised manuscript to clarify this point.

For cell quantification, eYFP+ cells were counted within a 500um² area of adventitia (for adventitial cells) or by counting the total number of eYFP+ cells within the media and intima of a single coronary artery (for medial and intimal cells) using the cell counter tool in ImageJ.

6) line 362, "?" can be deleted.

This has been deleted in the revised manuscript.

Referee #2:

Specific comments

1. The study focuses on proximal coronary artery, and there is no discussion on the heterogeneity of vascular smooth muscle cells. Indeed, VSMCs of proximal coronary arteries have been shown to derive from the neural crest, whereas more distal VSMCs are derived from epicardium (Jian et al. Development 2000). Performing neural crest-specific lineage tracing (e.g. Wnt1-Cre) would provide important insight into the lineage and cell-type origin of intimal fibroblasts of the proximal CoA.

This is an interesting point. Unfortunately, we cannot access Wnt1-Cre mice locally, meaning that this would be a major undertaking in terms of cost and time. Nevertheless, we hope to perform neural crest-specific lineage tracing in subsequent studies. We have discussed this point at length in the Discussion (see below) of the revised manuscript to convey the Referee's sentiment.

Thus, SMC play a critical role in driving vascular pathology in vasculitis. Intriguingly, SMCs can develop from multiple origins during embryogenesis (Wang et al., 2015). While those of the coronary artery are typically described as developing from epicardial precursors (Wang et al., 2015, Dettman et al., 1998), it is important to note that SMCs of the proximal coronary arteries – where vasculitis develops in KD – have been reported to instead develop from neural crest cells (Arima et al., 2012, Jiang et al., 2000). Indeed, our findings that relatively few SMCs of the proximal coronary arteries are labelled in *Wt1^{CreERT2}* system, supports a non-epicardial origin. Thus, while further studies are required with neural crest-specific lineage tracing, we predict that the luminal myofibroblasts that emerge during KD most likely develop from neural crest-derived SMCs. In this regard, it is intriguing to note that neural crest SMCs have enhanced collagen and proliferative responses to cytokines (compared to their mesoderm-derived counterparts) (Topouzis and Majesky, 1996, Gadson et al., 1997), raising the possibility that the susceptibility of the proximal coronary arteries to KD may be (at least partly) attributed to the unique potential of neural crest-derived SMCs.

2. Figure 1: The data in this figure is not particularly novel or useful, the model has already been described, and the fact that intimal fibroblasts express collagen is not surprising. This could be put into supplemental data. Furthermore, the usefulness of the FACS experiment is questionable considering the very low percentage of fibroblast labelled. Immunofluorescence quantifications are used throughout the manuscript, hence demonstration of co-labeling of fibroblasts in the areas of interest would be more appropriate. Finally, the choice of fibroblast marker is atypical, in particular for a manuscript focused on fibroblasts. The authors should use more robust markers such as PDGFRa, and use these in IF experiments.

As requested, we have moved Figure 1 to supplemental data (i.e. Expanded view 1) in the revised manuscript.

As suggested, we have changed the FACs plots to show PDGFR α expression (instead of Podoplanin) for the lineage tracing systems (Col1a2, Wt1CreERT2 and Myh11CreERT2). We have also performed co-labelling with immunofluorescence to analyse PDGFR α expression by the SMC-derived luminal myofibroblasts, which are the focus of this study (these data are shown in Fig. EV3 of the revised manuscript). Essentially, we find that eYFP+ cells labelled in the Col1a2 and Wt1CreERT2 systems express PDGFR α , consistent with labelling of fibroblasts. In contrast, the SMC-derived luminal myofibroblasts that emerge during vasculitis do not express PDGFR α and instead express mural cell markers, such as CD146. We have added the following section to the results section of the revised manuscript to describe these findings:

Phenotypic analysis of luminal myofibroblasts:

We next performed co-labelling studies to investigate the phenotype of luminal myofibroblasts. We first analysed cardiac sections from CAWS injected Myh11^{CreERT2} x R26^{eYFP} mice. This revealed that eYFP+ luminal myofibroblasts, and particularly those located near the lumen, expressed α -SMA and CD146, and had minimal PDGFR α expression (Fig. EV3A). We next investigated the phenotype of luminal myofibroblasts in humans by analysing autopsy tissue of two infants who died from myocardial infarction during acute KD. We found that in both KD cases, the inflamed coronary artery intima was heavily populated by α -SMA+ myofibroblasts that were uniformly CD146 positive (Fig. EV3B). Notably, α -SMA+ myofibroblasts of the adventitia were uniformly CD146 negative, exhibiting a distinct phenotype from their luminal counterparts (Fig. EV3B). These findings provide further evidence (in humans) that luminal myofibroblasts develop independently of adventitial myofibroblasts, and instead have a SMC origin.

3. Figures 2-3: Based on the images/data provided, one wonders whether the col1a2CreERT2 mouse would not be a better line for lineage tracing resident fibroblasts than Wt1CreERT2? Concerning Wt1CreERT2, early induction (E10.5) leads to epicardial labelling, and hence EPCD labelling in adult heart, including vascular smooth muscle cells and pericytes. This must be discussed. The fact that the authors see adventitial fibroblast labelling, but not VSMC labelling in proximal CoA again raises the issue of the origin of the VSMCs, that must at least be discussed in the manuscript.

As suggested, we tried lineage tracing in the Col1a2CreERT2 line (i.e. tamoxifen labelling before CAWS injection). However, in these experiments we saw relatively sparse eYFP labelling. We felt these data were not particularly helpful compared to the Wt.1CreERT2 system, which is the gold standard for labelling resident fibroblasts.

Regarding the minimal VSMC labelling in the WtCreERT2 system, we have elaborated on this point in Fig. EV2 and addressed the potential origin of VSMC (from neural crest) in the proximal coronary arteries in the revised discussion (see reply to point 1 of Referee above).

4. Figure EV1 also is not particularly informative. Authors state: "Moreover, RT-PCR analysis on sorted cardiac populations revealed that eYFP+ cardiac cells from Wt.1CreERT2/+xR26eYFP/+ mice express the highest levels of Col1a2 mRNA (Fig EV1), confirming their identity as resident fibroblasts". Does this mean the Podn+ cell population,

with lower *Coll1a2* expression, may not be fibroblasts? Co-labeling of *Wt1CreERT2*-lineage traced cells with fibroblast markers such as *PDGFR α* in sections is sufficient. What is the proportion of initial fibroblasts labelled?

To avoid confusion we have removed the RT-PCR data.

As suggested, we have performed co-labelling of *Wt1CreERT2*-lineage traced cells with the fibroblast markers *PDGFR α* by flow cytometry. These data are shown in **Fig. 1C** and **Fig. EV2A** of the revised manuscript. Furthermore, we have performed immunofluorescence to analyse *PDGFR α* expression on luminal myofibroblasts in **Fig. EV3A** of the revised manuscript. This population shows minimal *PDGFR α* expression as described in the following section of the revised results:

*We next performed co-labelling studies to investigate the phenotype of luminal myofibroblasts. We first analysed cardiac sections from CAWS injected *Myh11^{CreERT2} x R26^{eYFP}* mice. This revealed that *eYFP+* luminal myofibroblasts, and particularly those located near the lumen, expressed α -SMA and CD146, and had minimal *PDGFR α* expression (Fig. EV3A).*

5. Figure 8. The effect in the conditional knockout is striking. Could the authors check for lack of pS6 positivity and proliferation in VSMCs in the cKOs? What is the "infiltrate"? The authors should perform immunostaining e.g. CD45/*PDGFR α* ... to clearly establish whether or not these cells are fibroblasts.

We have confirmed that *eYFP* cells of *Myh11CreERT2 xRaptorfl/flxR26eYFP* mice (where SMCs have a conditional Raptor deletion) are pS6 negative. These data are included in **Fig. EV5A-B** and described (as below) in the results section of the revised manuscript:

*In control experiments, we confirmed that raptor deficient *eYFP+* SMCs were pS6 negative, illustrating that mTOR signalling is silenced in these cells (Fig. EV5A-B).*

We have also characterised the cells that drive residual stenosis in CAWS injected *Myh11CreERT2 xRaptorfl/flxR26eYFP* mice. We found that these cells are *eYFP* negative and α -SMA positive, suggesting these cells are myofibroblasts in which recombination has not occurred (due to lack of *cre* expression or inefficiency of tamoxifen), meaning that they still have intact mTOR. These findings are shown in **Fig EV5C** and described in the results section of the revised manuscript:

*Indeed, the residual stenosis that does develop in SMC-raptor deficient mice is driven by the infiltration of *eYFP* negative α -SMA+ cells (Fig. EV5C). The lack of *eYFP* expression indicates that these are α -SMA+ myofibroblasts where recombination (and hence raptor deletion) has not occurred, meaning mTOR is intact.*

Minor comments:

Check all insets e.g. Figure 4 F, the second inset in CAWS is not positioned correctly. Improve the appearance of the figures e.g. the illustrations representing mice are not very clear/useful.

Thank you, we have tightened the inserts and improved the clarity of the figures.

Referee #3:

Major comments:

1. Authors used WT1-CreERT2 with administration of tamoxifen at an early embryonic timepoint (E10.5) to label the embryonic epicardium and fibroblasts derived from this pool of cardiac progenitors. However, it is well known that, in addition to fibroblasts, epicardial progenitors also give rise to vascular smooth muscle cells in the coronary vasculature, so it is rather intriguing that authors do not see labeling of vSMCs by WT1-CreERT2.

This is a fair point, as SMCs of the coronary vasculature are typically described as having an epicardial origin. We have expanded our analysis to show that while *Wt1*CreERT2 recombination predominantly labels fibroblasts, a small proportion of vSMCs are also labelled (~9%) in this system. Why this is so low may relate to suboptimal recombination due to tamoxifen dose/timing. Alternatively, it is possible that cardiac vSMCs have multiple cellular sources. Indeed, as pointed out by Referee 2 (point 1), it has been reported that SMCs of the distal coronary arteries have an epicardial origin, while those of the proximal coronary arteries (which is the focus of this study) develop from neural crest cells. Please refer to our response to point 1 from Referee 2 which describes how we have elaborated upon this point in the discussion of the revised manuscript.

2. Authors should include in their discussion a sentence clearly stating that their evaluation of the contribution of pre-existing adventitial fibroblasts might be an underestimation, as the WT1-CreERT2 allele does not comprehensively label all cardiac fibroblasts. In addition to the epicardium, during cardiogenesis, fibroblasts can also be derived from other sources, such as the endocardium or, less often, neural crest (Moore-Morris, JCI 2014). Therefore, it is conceivable that pre-existing fibroblasts not labeled by WT1-CreERT2 might contribute to intimal fibroblasts.

We agree and have included the following statement in the discussion of the revised manuscript:

However, a potential caveat of our study is that the *Wt1*^{CreERT2} system does not label all resident fibroblasts, due to either suboptimal recombination (in *Wt1*+ epicardial cells) or the fact that fibroblasts can develop from alternate sources, such as the endocardium (Moore-Morris et al., 2014). Thus, it is possible that adventitial fibroblasts not labelled in *Wt1*^{CreERT2} system may form luminal myofibroblasts.

3. Authors provided compelling genetic evidence that mTOR is important for production of intimal fibroblasts from vSMCs. However, given the availability of mTOR inhibitors and the fact that the same group has reported that rapamycin prevents adverse vascular remodeling in the CAWS model, it would be interesting to assess the fate of Myh11-CreERT2 lineage-traced cells in mice receiving rapamycin treatment. Is this process as robustly blocked as it was in the conditional KO model?

Thank you, as suggested, we have performed this experiment and include the data as Fig. 7 of the revised manuscript. The findings are described (see below) in the results sections of the revised manuscript:

Finally, we explored the therapeutic potential of targeting mTORC1. To this end, we examined whether the pharmacological inhibition of mTOR signalling can block the formation of SMC-derived luminal myofibroblasts. To this end, tamoxifen treated *Myh11*^{CreERT2} x *R26*^{eYFP} mice were injected with CAWS to induce vasculitis and eight days

later, were treated continuously (3x/week) with either the mTOR inhibitor Rapamycin or vehicle control, until day 28 post CAWS (Fig. 7A). The analysis of cardiac sections revealed that eYFP+ SMC-derived cells had richly migrated into the inflamed coronary artery intima of vehicle-treated CAWS mice, causing profound arterial stenosis (Fig. 7B-C). In comparison, eYFP+ SMCs remained within the coronary artery media of rapamycin-treated mice, coinciding with minimal stenosis (Fig. 7B-C). These findings demonstrate that pharmacological mTOR inhibition can block the activation and/or migration of SMC-derived myofibroblasts during vasculitis, and in doing so, prevent pathological vascular remodelling.

4. The manuscript would benefit from a diagram summarizing the most relevant conclusions of the work in the last figure (somehow similar to a graphical abstract). This should highlight the contribution of vSMCs to intimal fibroblasts, but also the contribution of adventitial fibroblasts to all the reactive tissue that externally surrounds the media.

We thank the reviewer and have included a graphical abstract/synopsis in the resubmission.

Minor comments:

5. Line 105: "To report Cre expression, Col1a2CreERT2 mice were crossed to the R26-stop-eYFP mice to create Col1a2CreERT2 x R26eYFP mice". A reference to the manuscript originally describing the R26eYFP mice should be included here.

Thank you - done.

6. Microscopy images are mostly of very good quality, however, the low magnification stitchings are, sometimes, of lower quality, with a very strong grid pattern.

Images will be at a higher resolution in the final version, but unfortunately it may be difficult to fully eliminate the grid pattern from the images of larger areas, taken at a lower resolution.

Dear Dr. Stock,

Thank you for submitting your revised manuscript. It has now been seen by all of the original referees.

As you can see, the referees find that the study is significantly improved during revision and recommend publication. However, I need you to address the points below before I can accept the manuscript.

- Please address the minor concern of referee #1.
- Please rename the Conflict of Interest section as "Disclosure Statement and Competing Interests".
- We note that some authors have full stop after their first name in the manuscript file.
- As per our format requirements, in the reference list, citations should be listed in alphabetical order and then chronologically, with the authors' surnames and initials inverted; where there are more than 10 authors on a paper, 10 will be listed, followed by 'et al.'. Please see <https://www.embopress.org/page/journal/14693178/authorguide#referencesformat>
- The Author Checklist is currently missing the corr. author's name, ms ID# and the journal name (upper left corner).
- Funding information needs to be part of Acknowledgments. We note that funding information is currently missing in our manuscript tracking system eJP - i.e. Arthritis Rheumatology Australia Project Grant, the Reid Charitable Trusts, Victorian State Government Operational Infrastructure Support, Australian Centre for Transplant Excellence and Research
- We note that the following figure panels are currently not called out in the manuscript: Fig. 1H, Fig. 2B, Fig. 5C, Fig. 6F, Fig. 7D
- As per source data, we are unable to open the source data files for Figure 2H, 3C, 6 E, 6F, 6G, 7C, EV 1B, which seem like numerical data usually submitted in excel format. Please clarify.
- Our production/data editors have asked you to clarify several points in the figure legends:
 - o Please define the annotated p values */**/**/* as well as provide the exact p-values for the same in the legend of figure 1f; 2g-h; 6e-g; 7c-d; EV 1b; EV 4b; EV 5b; as appropriate.
 - o Please indicate the statistical test used for data analysis in the legends of figures 1f; 2g; 6e-g; 7c-d; EV 1b; EV 4b; EV 5b.
- The synopsis image needs to be 550px wide and 200-600px high. When your synopsis image is resized accordingly, the labels are too small to read (please see attached). Please provide a synopsis image with larger labels.

Thank you again for giving us to consider your manuscript for EMBO Reports, I look forward to your minor revision.

Kind regards,

Deniz Senyilmaz Tiebe

--

Deniz Senyilmaz Tiebe, PhD
Senior Scientific Editor
EMBO Reports

Referee #1:

The authors have thoroughly addressed all my comments, and I have no more questions for you. The study is highly interesting, clinically meaningful, well-written, and presented with a logical flow.

minor: line 369 : "Lactobacillus casei" should be italicized

Referee #2:

The authors have addressed the issues raised and substantially improved their manuscript. They have performed many of the additional experiments requested. Also, they have taken into account a number of important comments e.g. on the heterogeneity of VSMC origins, which has enabled them to greatly improve the text and interpretation of their results. Following the experiments with more robust fibroblast markers (PDGFR α notably), my feeling is that "lumal myofibroblasts" are more mural cell than myofibroblast, but there is no current consensus on this in the literature. Hence, I feel this manuscript, that addresses an important issue in vascular remodeling, is now suitable for publication.

Referee #3:

Authors have addressed my previous comments. In my opinion the manuscript deserves to be published. A minor edit is necessary, though: in Figure 1A-F and corresponding legend authors use "Wt.1" but this is incorrect as the gene name is "Wt1".

All editorial and formatting issues were resolved by the authors.

Dear Angus,

Thank you for submitting your revised manuscript. I have now looked at everything and all is fine. Therefore, I am very pleased to accept your manuscript for publication in EMBO Reports.

Congratulations on a nice work!

Kind regards,

Deniz

--

Deniz Senyilmaz Tiebe, PhD
Senior Scientific Editor
EMBO Reports

--
